# Dual enzyme-powered chemotactic cross β amyloid based functional nanomotors

Chandranath Ghosh[1,5], Souvik Ghosh[1,5], Ayan Chatterjee[1], Palash Bera[2], Dileep Mampallil[3], Pushpita Ghosh[4] & Dibyendu Das [1] ✉

Nanomotor chassis constructed from biological precursors and powered by biocatalytic transformations can offer important applications in the future, specifically in emergent biomedical techniques. Herein, cross β amyloid peptide-based nanomotors (amylobots) were prepared from short amyloid peptides. Owing to their remarkable binding capabilities, these soft constructs are able to host dedicated enzymes to catalyze orthogonal substrates for motility and navigation. Urease helps in powering the self-diffusiophoretic motion, while cytochrome C helps in providing navigation control. Supported by the simulation model, the design principle demonstrates the utilization of two distinct transport behaviours for two different types of enzymes, firstly enhanced diffusivity of urease with increasing fuel (urea) concentration and secondly, chemotactic motility of cytochrome C towards its substrate (pyrogallol). Dual catalytic engines allow the amylobots to be utilized for enhanced catalysis in organic solvent and can thus complement the technological applications of enzymes.

Mimicking out-of-equilibrium processes of Nature for material actuation can help in the design of active and adaptive systems with lifelike dynamic properties. Biological machines like enzymes or walker proteins harness energy from catalytic reactions to spatiotemporally regulate various critical functions such as signal-driven intracellular chemical transport, cell motility and division, muscular contractions, and endo/exocytosis[1,2]. Among these, specifically, the involvement of biological machines for an intriguing organismal feature like chemotaxis (directional motion in response to chemical gradient), plays important roles in survival[3], immune sensing[4], and fertilization of ova[5]. In experimental setups, biological machines like enzymes have shown enhanced diffusive motion via actuation of force from substrate turnover, along with directed stochastic motion towards substrate gradient[6–10]. Further, orthogonal processes are utilized for motility and chemotaxis in cell migration[11–13]. More specifically, while motility is regulated by the consumption of energy, directionality is controlled by sensing of higher substrate gradient by a spatial sensing mechanism. Incorporating such dynamicity into synthetic support via harvesting energy can lead to the realization of advanced functional materials that are capable of showing autonomous directional motility for multifarious tasks[14–25]. Importantly, such division of labour (for example, motility and directionality) by integrating the different yet interrelated processes is a critical challenge in the field of self-powered biomimetic constructs. In this context, biologically derived precursors like peptides and nucleic acid-based supramolecular frameworks are interesting candidates for nanomotor chassis (framework of the assemblies) to attach propulsive units (bio/chemical catalyst)[26–31]. Significantly, this could open up new possibilities in the emerging field of soft-nanomotors, especially from the context of biocompatible platforms, and subsequently, could be exploited in a wide range of theranostic/biomedical applications such as biosensing, drug delivery, and microsurgeries[26–31].

[1]Department of Chemical Sciences and Centre for Advanced Functional Materials, Indian Institute of Science Education and Research (IISER), Kolkata, Mohanpur 741246, India. [2]Tata Institute of Fundamental Research (TIFR), Hyderabad, Telangana 500046, India. [3]Department of Physics, Indian Institute of Science Education and Research (IISER) Tirupati, Mangalam, Andhra Pradesh 517507, India. [4]School of Chemistry, Indian Institute of Science Education and Research (IISER), Thiruvananthapuram, Kerala 695551, India. [5]These authors contributed equally: Chandranath Ghosh, Souvik Ghosh. ✉e-mail: dasd@iiserkol.ac.in

Peptide assemblies being biocompatible and biodegradable, are synthetically amenable via specific mutations that can have profound effects on morphology and function[32–39]. However, the major hurdle of using these as nanomotor chassis stems from their dynamicity and inherent plasticity that result in a lack of robustness with lesser persistent lengths[40,41]. To circumvent structural deformity which may occur due to opposing viscous drag forces in the low Reynold's number regime, peptide assemblies should feature streamlined morphologies with longer persistent lengths[42]. Oligomeric fragments of Aβ (1–42) peptide sequence, seen as protein deposits in Alzheimer's disease, could be used due to its propensity to access robust nanotubular morphologies[43–51]. Further, its multi-enzyme loading capabilities can help the nanoconstruct to explore surrounding environments with directional control (Fig. 1). In this context, advanced extant biological events such as cell migration exploit two orthogonal processes for motility and directionality[11–13]. It would be intriguing to develop amyloid-based synthetic soft nanoconstructs that can exploit orthogonal reactions for different functions. Towards this end, we used paracrystalline nanotubular amyloid morphologies as nanomotors (amylobots) chassis and installed two enzymes, one for active motion (amidohydrolase, urease) and the other for navigation control (peroxidase, cytochrome C, CytC)[49–52]. The navigation control coupled motility led to the development of a chemomotile soft nanoconstruct that is used as a platform for augmented peroxidase activity, both in an aqueous and non-aqueous milieu.

## Results

### Selection of peptide sequence

We started with a short peptide sequence, Ac-KLVFFAL (Ac-KL) from the nucleating core of wild type Aβ (1–42)[49–51]. Upon assembly, Ac-KL accessed well-defined nanotubular morphologies (diameter = 32 ± 2 nm, height = 10 ± 1 nm, and length = 5–20 μm), (TEM, SEM, AFM,

Fig. 2a–c). CD and FTIR further provided insights of the β-sheet signature and packing of Ac-KL (Supplementary Fig. 1)[32,40–44]. We predicted that the large aspect ratios of the longer nanotubes would hinder their facile motility due to higher viscous drag. Mechanical force was utilized via probe sonication for ca. 30 min which shortened the nanotubes[50]. TEM, SEM and AFM suggested retention of morphologies with length variation from 180–560 nm (Fig. 2d–f, Supplementary Figs. 2 and 3). Further, binding studies done with negatively charged gold nanoparticles suggested the presence of solvent-exposed cationic lysine residues (Fig. 2g, j, Supplementary Fig. 4, Supplementary Information)[35,48]. The amphiphilic binding surfaces of the nanotubes created from antiparallel orientations of peptide strands could also bind to hydrophobic dyes like Nile red (CLSM, Supplementary Fig. 5)[51].

### Introduction of motility

To install motility, urease was chosen as the bioengine for its capability of efficient chemo-mechanical energy conversion by converting urea to ammonia and $CO_2$[52–54]. Urease converts neutral urea to oppositely charged ionic species and generates local electric field due to the difference in diffusivity of the ions which finally causes ionic self-diffusiophoresis[6]. The negatively charged urease (pI 5.1) bound to the short (sonicated for 30 min) and long cationic nanotubes (no sonication) with loadings of $10.82 ± 2.5$ μg nmol$^{-1}$ and $11.01 ± 1.7$ μg nmol$^{-1}$ respectively (TEM in Fig. 2h, k, the concentration of exposed urease was 15 μM). RITC-tagged urease was used to probe the localization of protein on the nanotube surface (CLSM, Fig. 2i, l). To check the propulsion under time-lapse microscopy, the urease-loaded amylobots were mixed with varying concentrations of urea (Fig. 3a)[55,56]. With the increase of urea, a significant increase in motility was observed (Supplementary Movies 1–5, Fig. 3b, control without urea in 3c). Trajectories of the short nanotubes demonstrated a fuel concentration-

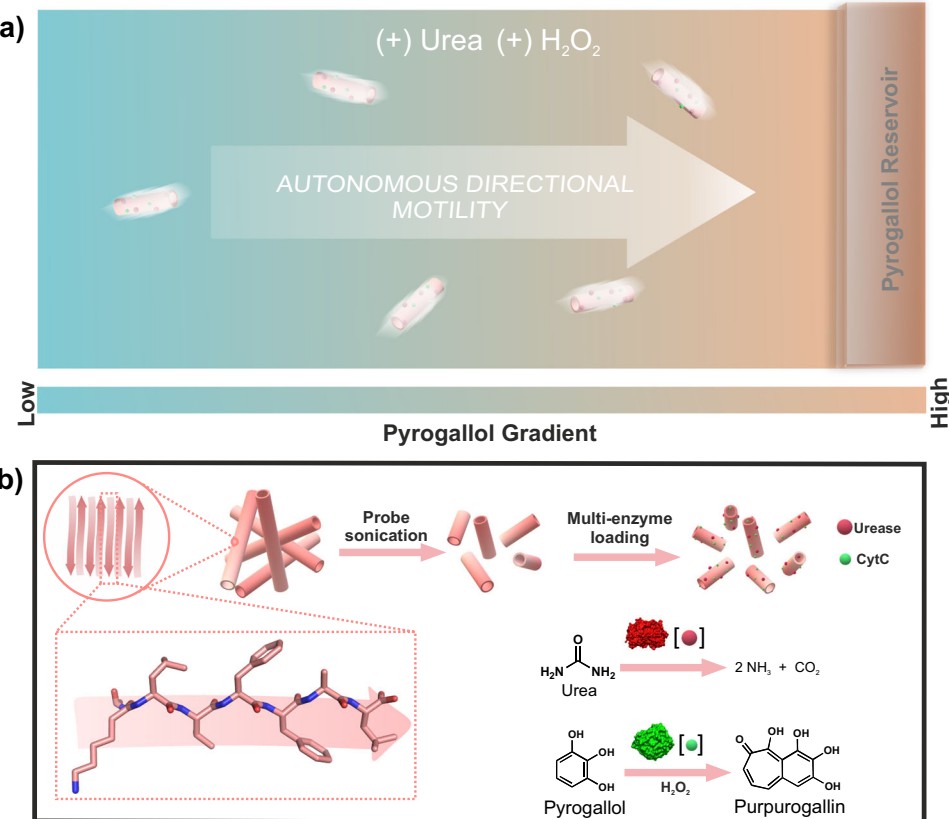

**Fig. 1 | Schematic representation of chemotactic motility of amylobots. a** Representative image of cytochrome C bound urease powered amylobots towards pyrogallol reservoir. **b** Illustration of step-wise design of amylobots via probe sonication of long nanotubes followed by binding with enzymes.

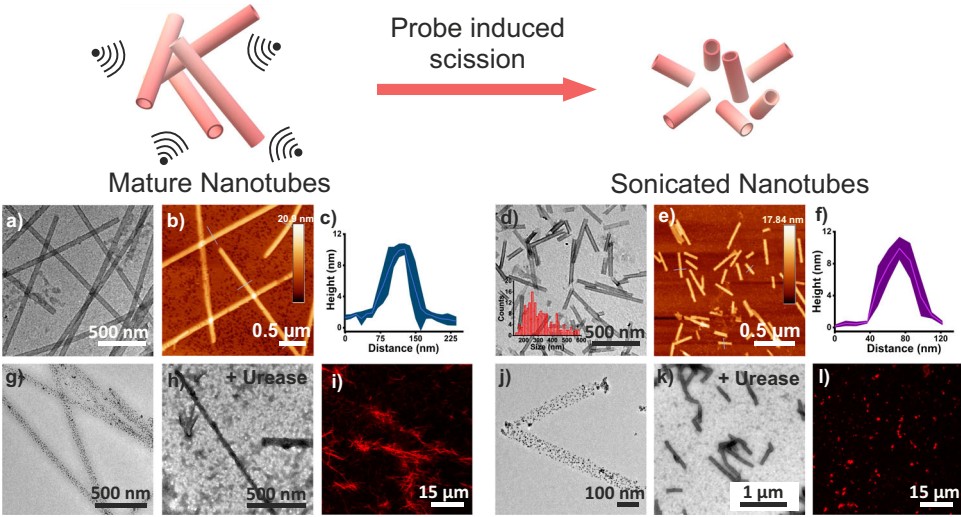

**Fig. 2 | Illustration depicting sonication-induced shortening and characterization of nanotubes. a–c** TEM, AFM and height profiles (AFM) of mature Ac-KL nanotubes. For AFM, the error band represents line analyses of three different nanotubes. Data are presented as the mean ± s.d. ($n = 3$). **d–f** TEM, AFM and height profiles (AFM) of short nanotubes respectively. The error band of AFM represents line analyses of three different nanotubes. Data are presented as the mean ± s.d. ($n = 3$). **g** TEM of gold nanoparticles bound long nanotubes. **j** TEM of gold nanoparticles bound shortened nanotubes. TEM and CLSM of urease bound **h, i** long and **k, l** sonicated nanotubes respectively (RITC tagged urease used for CLSM). All the experiments were repeated for at least three times.

dependent trend (Fig. 3d). Mean square displacement (MSD) was calculated from the trajectories to yield respective velocities which also showed a urea-dependent increasing trend (Fig. 3e, inset, 20 particles were tracked, Supplementary Fig. 6)[55,56]. This suggested that the Brownian motion was overcome by the force actuated via urease catalysis during substrate turnover[57]. Long nanotubes showed insignificant active motion with urea as expected from the higher viscous drag (Supplementary Movie 6). Notably, nanotube length variation studies were also done with different sonication times, and the active motion was subsequently analyzed (Supplementary Fig. 7, Supplementary Information)[58]. Further, the effect of enzyme loading on the motility of the amylobots was analyzed by exposing the varying concentrations of the enzyme to the sonicated nanotubes (Supplementary Fig. 8, Supplementary Information). Nanotubes sonicated for 30 min were used for further studies in this work.

For better understanding, an individual-based/particle-based model was proposed where each amylobot was described as a cylindrical nanotubular particle with the aforementioned dimensions (Supplementary Information)[59,60]. The equations of motion of an individual nanotube were considered to follow dynamics similar to an overdamped Brownian motion. The model assumed that each nanotube experiences a repulsive mechanical force ($F_{rf}$) while interacting with the neighbouring nanotubes, a random force (G) from the surroundings, and a self-propulsion force ($F_{mf}$, Supplementary Fig. 9). The motility force stemmed from the urease catalysis. Two-dimensional trajectories were obtained when the simulation was done for different values of motility forces starting from $f_u = (0-24) \times 10^5$ Pa μm² (Fig. 3f). Defining a quantity $f_{mot} = 1 \times 10^5$, $f_u$ values have been expressed in terms of $f_{mot}$ throughout the text. Furthermore, MSD was calculated as a function of the lag time (Fig. 3g), to determine respective velocities. Without any motility force, motion resembled passive Brownian-like motion whereas, in the presence of motility forces, active motion and enhanced dispersion are reflected in the particle trajectories and their MSDs (Supplementary Movies 7–10) were in excellent agreement with the experimental results.

This simulation model not only captured the experimental observations in terms of the individual motion characteristics of the particles but also predicted the probable velocity for different urea concentrations (75 mM, Supplementary Fig. 10). Further, to get better insight into the spatiotemporal dynamics of amylobots, the MSD

values were fitted in the equation MSD = $4D\Delta t^\alpha$, where $\Delta t$ is the time interval, $D$ is the diffusion constant and $\alpha$ is the MSD exponent. The superdiffusion of the amylobots was found to be increased with the increase of the urea concentration (Supplementary Fig. 11, see Supplementary Information for details)[61,62].

## Chemotaxis of amylobots by installation of a navigational handle

To install long-range chemotactic motility, CytC, a peroxidase with orthogonal substrate specificity with respect to the existing urease (amidohydrolase) was co-localized on the surface of the amylobots. The enzyme with higher catalyzing efficiency (urease) would be the primary propulsive engine and contribute to the active motility whereas the CytC with lower efficiency might be responsible for chemotactic control towards a specific substrate gradient[8,11–13,16]. The co-localization of urease and CytC was confirmed by CLSM using RITC and FITC-tagged proteins (Fig. 3h–j). From AFM and TEM investigations different patch-like globular structures were observed on the nanotube surface suggesting the local inhomogeneity in enzyme distribution (Supplementary Fig. 12, Supplementary Information)[63]. To investigate the chemotactic behaviour of the dual enzyme-loaded amylobots, a homemade setup was designed to monitor the relative population of the urease-CytC-bound nanotube (Fig. 4a)[64–66]. The population of the urease-CytC-bound nanotube was always found to be significantly higher near the pyrogallol reservoir, thus suggesting the chemotactic migrations of the amylobots (at point 3, Fig. 4b, Supplementary Fig. 13).

Notably, different control systems with varying conditions were unable to display the chemotactic migration of the nanomotors (Supplementary Figs. 14–18). Firstly, to show the combined effect of the CytC and the pyrogallol gradient, control experiments were performed excluding one of the components at a time from the system. When the experiment was performed with only urease-loaded nanotubes (CytC was absent but pyrogallol gradient was present), expectedly a random trend in the change of population of the nanotubes was observed (Supplementary Fig. 14). Also, in the absence of a pyrogallol gradient (the reservoir 'B' was filled with buffer instead of pyrogallol), a similar observation with no distributional bias was seen (Supplementary Fig. 15). Both results strongly suggest the role of the additional enzyme (CytC) for the observed chemotactic migration. In addition,

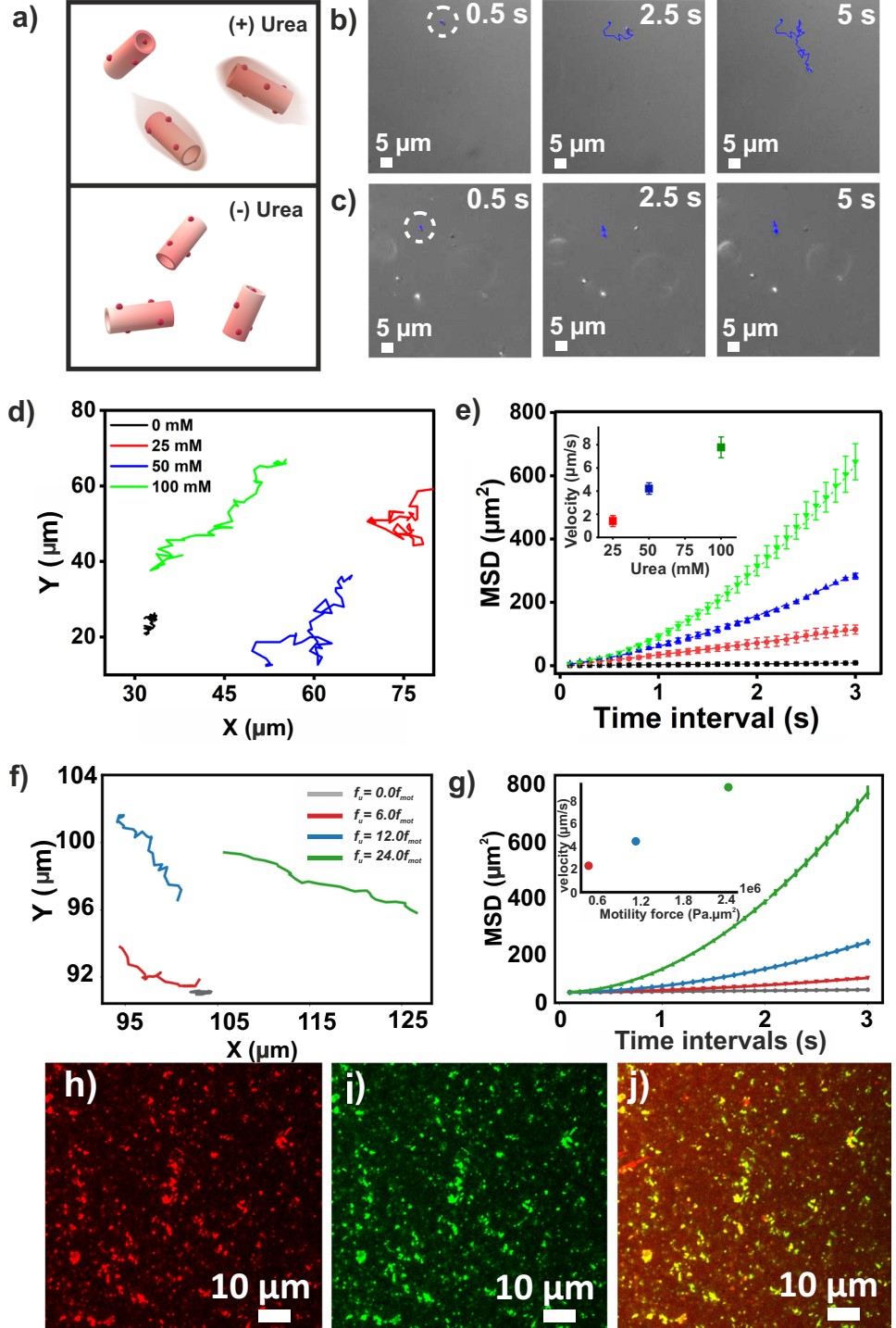

**Fig. 3 | Design and analysis of the motile amylobots. a** Schematic illustration and representative snapshots of amylobots in **b** the presence and **c** absence of urea vs time. **d** Tracking trajectories and **e** MSD vs time interval with varying urea concentration (inset shows velocity profile). Data are presented as the mean ± s.d. (*n* = 20). **f** Simulated trajectories of a nanotube with different values of motility force ($f_u$). **g** Ensemble averaged simulated MSD as a function of time for different motility forces. Data are presented as the mean ± s.d. (*n* = 20). Inset depicts velocity variation with motility force/urea concentration. CLSM images of amylobots bound with **h** RITC-urease, **i** FITC-CytC and **j** merged image.

freely diffusing CytC was added in the medium along with only urease-loaded nanotubes to probe the possibility of whether the non-specific binding of the product generated from the catalytic activity could show the directionality of the nanotubes towards the pyrogallol gradient. However, no specific increment in the population of the nanotubes was observed (Supplementary Fig. 16). Further, the agarose gel was allowed to release the pyrogallol for a longer time period (ca. 8 h)

which would lead to equilibrated pyrogallol concentrations with a homogenous distribution. In this case, as well, the population of nanotubes at points 3 and point 1 failed to show any particular bias (Supplementary Fig. 17). To rule out any unexpected effects from the combination of reactions (peroxidase by CytC with pyrogallol and the urea-urease reaction), the gel was mixed with free CytC and pyrogallol was added at the reservoir and the system was equilibrated (ca. 1 h)

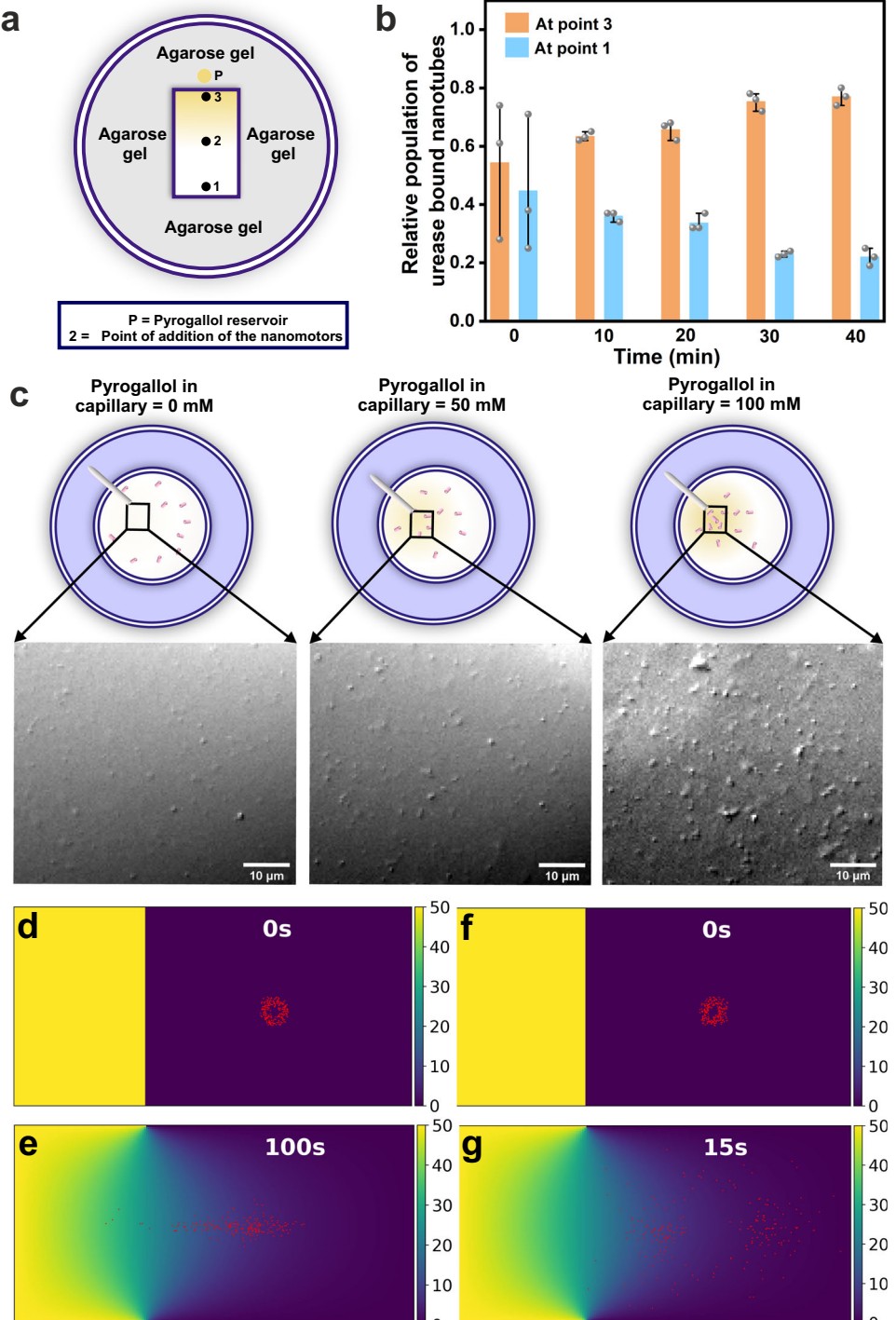

**Fig. 4 | Analysis of the chemotactic migration of the amylobots.** Schematic representation of **a** the experimental setup and **b** time-dependent populations of the nanomotors at different positions. The error bars were calculated from three separate experiments. Data are presented as the mean ± s.d. (*n* = 3). **c** Experimental setup and DIC micrographs near the opening of the capillaries loaded with pyrogallol after 15 min. Simulated time snaps of the spatial profiles of **d**, **e** urease-CytC loaded nanotubes and **f**, **g** only urease loaded nanotubes with time.

before the addition of the nanomotors (only urease loaded nanotubes) in the medium (at point 2). Again, no distribution bias in the population could be observed eliminating any cumulative effects generated from the orthogonal reactions (Supplementary Fig. 18). To eliminate the possibilities of convective flow generated from different sources, all possible measures were taken, several controls were done that are listed below, and all the experiments were done multiple times. To prevent air-induced convection the petri plate was covered while

performing the experiments. Further, all the experiments were performed at room temperature (temperature variation of less than ±1 °C). The temperature variation in the local aqueous environment for the catalytic reactions by urea-urease and CytC-pyrogallol is expected to be negligible. The possibility of any drift flow on nanotubes that could have occurred by the chemical cue released from the agarose gel was ruled out by performing control experiments with another enzyme (GOx, Supplementary Fig. 19, see Supplementary Information

for details). Instead of CytC, in this case, GOx was loaded along with urease. However, no directional bias was observed towards the pyrogallol gradient (Supplementary Fig. 19, 'P' stands for pyrogallol in the reservoir). This suggested that uneven substrate distribution, density gradient caused by chemicals, or probable osmotic flows originating from pyrogallol gradient did not induce any collective migration.

The chemotactic migration was further supported by the time-dependent population measurement of urease-CytC-bound nanotubes (loaded along with RITC tagged urease in 9:1 molar ratio) through fluorescence assay (Supplementary Fig. 22, Supplementary Information). In the absence of urea, the amylobots loaded with urease-CytC did not show chemotaxis suggesting that bound CytC alone was unable to move the motors towards the chemical cue (Supplementary Figs. 20 and 22b). However, CytC bound on nanotubes (in the absence of urease) showed enhanced diffusion when the pyrogallol concentration was varied in a homogeneous medium (Supplementary Fig. 21, Supplementary Movie 11). Hence, although the primary power source is derived from the urease, the chemotactic ability originated from the enhanced diffusivity of CytC. Notably, rapid generation of purpurogallin was observed in the gel buffer interface with a 7-fold higher rate than the controls without urea (Supplementary Figs. 20 and 23)[46]. This result suggested that the chemotaxis towards higher pyrogallol concentrations at the interphase led to the higher catalytic activity of CytC (as substrate concentration was more). While in the absence of urea, the nanotubes showed a random distribution of motion and did not lead to specific localization to higher pyrogallol concentration (gel-buffer interface) leading to lower catalytic rates of CytC (as pyrogallol concentration was low). The directional motility was further confirmed from the microscopic observations (Fig. 4c)[67–70]. Different concentrations of pyrogallol (0, 50 , 100 mM) loaded glass capillaries (sealed at one end) were suspended in the buffer solution containing urease and CytC-loaded nanomotors and $H_2O_2$ in a 35 mm glass-bottom petri plate. After ca. 15 min, the population at the capillary opening was found to be increased for higher concentrations of pyrogallol (Fig. 4c, Supplementary Fig. 24). However, no specific localization of the nanomotors near the capillary opening was observed in the absence of pyrogallol (with only buffer). This observation strongly underpinned the role of pyrogallol (chemical cue) and CytC for directional motility. The chemotaxis was further observed under confocal microscopy when similar experiment was performed with the RITC-tagged urease (Supplementary Fig. 25, see Supplementary Information for details)[24]. Furthermore, the motion of the dual enzyme-loaded nanotubes was characterized by varying pyrogallol concentrations. MSD and the velocity were calculated from the trajectory analysis (Supplementary Fig. 26). The chemotactic ability of only CytC-loaded nanotubes was investigated using a similar setup in the absence of urea. We monitored the population at the opening of the capillary. A modest increase was observed after a prolonged time of 1.5 h in the overall population of the only CytC-loaded nanotubes (Supplementary Fig. 27a, b, a similar observation was observed for the control experiment done in the absence of urea with urease-CytC-nanotubes, Supplementary Fig. 27a, c). It would be important to mention here that in presence of urea, the urease-CytC nanotubes showed localization in ca. 15–20 min. In combination, these results suggest that motors in the absence of urea do have a subdued yet finite chemotactic propensity towards a gradient of pyrogallol and it takes a longer time to achieve this. Further, the diffusivities calculated for only CytC-loaded nanotubes in the absence of urea/urease (at 100 mM pyrogallol) were found to be significantly lesser compared to urease-loaded nanotubes ($D_{urease/cytc} = 6.99 \pm 0.051$ μm$^2$/s, $D_{cytc} = 0.42 \pm 0.017$ μm$^2$/s).

For distinguishing the active (urease and CytC-loaded, showing chemotaxis) and passive nanotubes (only urease-loaded), further control experiments were performed where both the nanotubes were mixed, and motility was monitored microscopically in the presence of homogenous urea. Briefly, a similar setup (35 mm glass bottom petri plate) was used for this (Supplementary Fig. 28). A capillary filled with pyrogallol (sealed at one end) was suspended and a mixed system of active and passive nanomotors was dispersed in the medium (containing uniformly distributed urea and $H_2O_2$). For differentiating the systems, active and passive nanotubes were loaded with FITC and RITC tagged urease (loaded along with untagged urease in 1:9 molar ratio) respectively. The fluorescence intensity (FI) of the fluorophore-labelled nanotubes was monitored at the capillary opening using the confocal microscope. The normalized fluorescence intensity of FITC-labelled nanotubes (active ones) was found to be gradually increased with time at the opening of the capillary mouth, whereas for RITC-labelled nanotubes no specific trend was there (passive ones, Supplementary Fig. 28). This observation clearly suggested that only the urease-CytC loaded nanotubes demonstrated the capability to exhibit chemotaxis towards pyrogallol gradient.

To explore the spatiotemporal dynamics of the dual-enzyme-loaded amylobots, a unified scheme was followed for the simulation model. The substrate was modelled as a diffusive field at one end of a rectangular simulation box. Upon circular inoculation of the amylobots, the motors exhibited directed motion towards the pyrogallol reservoir (Fig. 4d, e, Supplementary Movie 12). The motility forces corresponding to urease and CytC are taken as $f_u = 12 \times 10^5$ Pa μm$^2$ and $f_c = 10$ Pa μm$^2$, where $f_u \gg f_c$. To probe the individual roles of the enzymes, control simulations in the absence of urea ($f_u = 0$) demonstrated the usual Brownian motion (Supplementary Fig. 29, Supplementary Movie 13). Simulated MSD for chemotactic motion (in the presence of urea) showed an initial linear fit followed by a parabolic trend suggesting the time required to sense the concentration gradient (Supplementary Fig. 30). However, in the absence of urea, the MSD plot showed a line fit suggesting Brownian-type motion (Supplementary Fig. 31). Furthermore, the control simulation in absence of CytC did not show any directional motion (Fig. 4f, g, Supplementary Fig. 32 shows the computationally derived rotational MSD, see Supplementary Information for details). Thus, the model supports the spatiotemporal motion of amylobots by capturing all the different situations in line with the experiments.

## Proposed mechanism of chemotactic motility

We have proposed the following mechanism for the chemotaxis seen in the case of the dual enzyme-loaded nanotubes in a gradient of pyrogallol. For motility, urease uses chemo-mechanical energy conversion via the generation of a local electric field from the difference in diffusivities of the oppositely charged ionic species (as shown by many reports)[6,58]. Hence, when urea was uniformly distributed in the medium (and no CytC + Pyrogallol gradient), the urease-bound nanotubes only showed the persistent Brownian motion without any specific bias towards any direction. The persistent Brownian motion indicated spatial asymmetry in the urease reaction, which was presumably due to the inhomogeneous binding of the enzymes on the rod-like nanomotor chassis (inhomogeneous binding could also be seen from AFM images, Supplementary Fig. 12)[57]. However, the persistent motion of the nanomotors did not create any directionality at long durations due to the rotational Brownian motion[71]. It is expected that constant random rotation was due to the asymmetry in the catalytic reaction (as local concentration fluctuations can produce a transient asymmetry)[71]. Thus, the nanomotors had persistent motion but without any specific bias in the directionality of the overall motion as the substrate urea was homogeneously distributed.

Now to rationalize the chemotaxis, we propose that an additional mechanism leads to a decline in this random rotational Brownian motion of the nanomotors. In this context, it is known that the chemotaxis of Janus particles involves their active rotation which is coupled with the direction of fuel gradient[8,72]. A detailed

theoretical derivation of this active rotational component can also be found in the literature[73]. We propose a similar mechanism in our system in the inhomogeneous presence of an additional weak enzyme (CytC). The random rotation of the nanomotor chassis diminishes presumably due to the CytC-induced active rotations towards the pyrogallol gradient despite the weak catalytic proficiency. This installs a weak external field which eventually restricts the random rotations of the urease-CytC-powered self-diffusiophoretic motors[8]. In other words, this additional interaction (CytC-pyrogallol) disrupts the overall randomness in directionality in the strong persistent motion powered by urease, and subsequently, the motors now make a gradual choice towards the external bias of the gradient of the CytC-substrate[71,74]. However, after rotation, there would still be an equal possibility of the motors to go against or towards the gradient. We argue that the choice towards the pyrogallol gradient could partially be attributed to the weak yet finite diffusivity of the CytC (CytC-loaded nanotubes in the absence of urease indeed showed enhanced diffusivity at 100 mM pyrogallol, although the diffusivity was significantly lesser compared to urease-CytC loaded nanotubes, $D_{\text{urease/cytc}} \gg D_{\text{cytc}}$ in buffer). However, understanding the interplay between the orthogonal enzymes, that break the symmetry of equal possibility of moving against or towards the pyrogallol gradient, requires further dedicated investigations and our current efforts are towards this front.

## Enhancement of enzyme activity in organic solvent

Activating enzymes in harsh organic solvents has applications in pharmaceutical and specialty chemical industries. Inter-particle diffusion limitations are encountered in biphasic enzymatic reactions due to poor substrate accessibility between heterogeneous phases. We argued that the chemotactic motility can be useful to overcome diffusion limitation in a biphasic environment of buffer and organic solvent and can lead to higher catalytic rates. As a proof of concept, a small percentage of buffered samples was added to toluene (chosen for its comparable logP value (2.678), with purpurogallin, logP (2.416), logP of substrate pyrogallol (0.294) is significantly lower due to its hydrophilic nature, details in Supplementary Information)[46,75]. Microscopic motility towards the substrate-rich region was expected to improve the CytC activity. Amylobots were mixed with different concentrations of urea, toluene, pyrogallol, and $H_2O_2$ were added to a vessel (Fig. 5a, Supplementary Information). The activity towards rapid purpurogallin generation could be observed with rates directly proportional to the urea concentration (Fig. 5b). In presence of urea, amylobots showed 9-fold higher peroxidase activity ($v_i = 35.1 \pm 7.02$ μM min⁻¹) compared to control ($v_i = 3.9 \pm 1.03$ μM min⁻¹, Fig. 5b). Further, this activity was 2-fold higher than the native activity of CytC in the buffer. The catalytic rate enhancement of enzymes in organic solvents by the functionalized amylobots could be adaptable in future for chemical industries that use enzymes in non-aqueous heterogeneous environments.

## Discussion

In conclusion, the work demonstrated the development of cross β amyloid-based nanomotor chassis with chemotactic motility using two different bio-engines. The binding capabilities of these soft nanoconstructs were exploited to host dedicated enzymes for active motion (amidohydrolase, urease) and the other for navigation control (peroxidase, cytochrome C). Significantly, two distinct transport behaviour via these two orthogonal enzymes were utilized, where enhanced diffusivity of urease with increasing fuel (urea) concentrations was used for active motion while CytC was used for chemotactic migration towards the chemical cue (pyrogallol). This system thus mimicked the advanced extant biological events such as cell migration where two orthogonal processes are used for motility and directionality. Notably, a particle-based simulation model was used to further support the experimental data. Further, the super-diffusive motion by the urease and chemical cue sensing from CytC, facilitated substrates accessibility and subsequently led to rate enhancements in a biphasic milieu. Importantly, this two-enzyme-regulated chemotactic movement shown by the synthetically accessed soft nanoconstructs further signifies the importance of the distribution of the payload observed in the contemporary motile biological systems. The bio-friendly chassis alongside the mutualistic relationship between rationally chosen bio engines can be useful in the future as a smart catalytic paradigm with industrial implications.

## Methods

Fluorenylmethyloxycarbonyl (Fmoc) protected amino acids, piperidine, activator N,N′-diisopropylcarbodiimide (DIC), trifluoroacetic acid (TFA), hexafluoro isopropanol (HFIP), gold chloride trihydrate, sodium borohydride and trisodium citrate dehydrate, agarose, Bradford reagent, rhodamine B isothiocyanate (RITC), fluorescein isothiocyanate isomer I (FITC) and Nile red were purchased from Sigma Aldrich Merck. Oxyma was purchased from Nova biochem. 4-(2-hydroxyethyl)−1-piperazineethanesulfonic acid (HEPES), urease, urea, pyrogallol was purchased from TCI, Japan. Cytochrome C (oxidized) extra pure and triethylsilane were purchased from SRL and Spectrochem respectively. Hydrogen peroxide (30% w/v solution), Fmoc-rink amide MBHA resin, and all solvents were purchased from Merck. Milli-Q water was used for all the experiments.

### Peptide synthesis

Peptide (Ac-KLVFFAL-CONH₂) synthesis was performed in Aapptec peptide synthesizer. Firstly, Fmoc-rink amide MBHA resin of a loading capacity of 0.5 mmol/g, was swollen for 15 min in DMF. Next, deprotection of Fmoc group was carried out with 20% piperidine in DMF. The coupling step of every amino acid was performed using DIC and the oxyma solution in DMF. Further, N-terminus lysine of the peptide sequence was acetylated using acetic anhydride in DMF. Lastly, the resin was washed with DCM and DMF and air-dried. After that, peptide cleavage from the resin was performed with using of TFA / triethylsilane (5:0.1 v/v) at room temperature for 2 h. TFA was removed

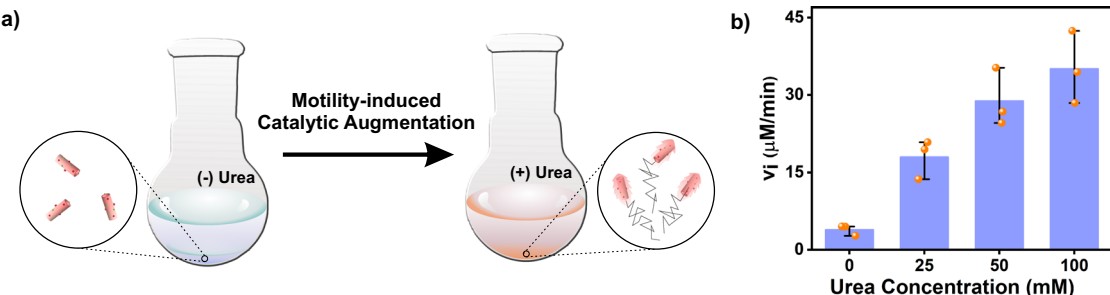

**Fig. 5 | Chemotactic motility induced catalytic augmentation. a** Schematic representation of the experimental design. **b** Initial rates of oxidation vs urea concentration. Data are presented as the mean ± s.d. ($n = 3$ independent experiments).

by using high vacuum from cleaved peptide-TFA solution. Peptides were precipitated in ice-cold diethyl ether with dropwise addition followed by centrifugation at 6000 rpm for 30 min at 4 °C in Eppendorf centrifuge 5804R leading to white precipitate. The white precipitate was washed thrice using cold diethyl ether. Purification of crude peptide was done by using Atlantis T3 C18 preparative reverse phase column in preparative Waters HPLC system with a linear gradient of water containing 0.1% TFA and acetonitrile containing 0.1% TFA. Finally, the molecular weight was confirmed by Bruker Mass Spec Q-tof systems.

Ac-KLVFFAL-CONH$_2$ (Ac-KL) (C$_{46}$H$_{71}$N$_9$O$_8$) (*m/z*) calculated for [M + H$^+$]: 878.54; found: 878.55. ESI: 878.55.

### Negatively charged gold nanoparticle synthesis and binding study

Briefly, freshly prepared 15 μL of 1 mM trisodium citrate solution was added to a round bottom flask containing 3.82 mL of Milli-Q water. Further, 18 μL of 83.3 mM of HAuCl$_4$ was added to it in stirring condition. To this mixture, 100 μL of ice cold, freshly prepared solution of NaBH$_4$ (100 mM) was added gradually and continued stirring for one hour. Over time the solution turned pink, thus indicating gold nanoparticle formation. Moreover, a localized surface plasmon resonance (SPR) transition was observed at 517 nm (Supplementary Fig. 4a), which suggested the formation of gold nanoparticles. From transmission electron microscopy (TEM), the average particle diameter was found to be 5–15 nm (Supplementary Fig. 4b).

For gold nanoparticle binding studies, 5 μL aqueous dispersions of the short Ac-KLVFFAL-CONH$_2$ nanotubes were taken and 200 μL of gold colloid (−AuNP, 0.3 mM) was added to it. The mixed solution was kept for 4 h for incubation at room temperature. Once a precipitate of purple-red colour was formed the mixture was centrifuged. The pellet formed at the bottom of the microcentrifuge tube was redispersed in water. To check the binding microscopically, TEM was done. Ten microlitres of aliquots was pipetted out and drop casted on the TEM grid and kept for 2 min. Then the excess solvent was soaked up using filter paper.

### Synthesis of RITC labelled urease and FITC labelled CytC

RITC (0.4 mL, 5 mg/mL in DMSO) was added dropwise into 4 mL of Na$_2$CO$_3$–NaHCO$_3$ buffer (pH = 9.0, 50 mM) containing 20 mg of urease. The mixture was stirred for 4 h at room temperature followed by neutralization with NH$_4$Cl (50 mM). The solution was dialyzed (12 KDa cellulose tubing) in a buffer (pH 7.0, PBS, 50 mM) for 24 h to eliminate the unreacted RITC[76].

For tagging CytC with FITC, the protocol provided in the FluoroTagTM Conjugation Kit (Product no. FITC1, Sigma Aldrich) was followed. Briefly, 1 mL of FITC (1 mg/mL stock in 0.1 M Na$_2$CO$_3$–NaHCO$_3$ buffer, pH = 9) was added to 4 mL of CytC (5 mg/mL stock in 0.1 M Na$_2$CO$_3$-NaHCO$_3$ buffer, pH = 9) with continuous stirring for 2 h under dark. The labelled protein was then purified by G-25 M Sephadex column using PBS and was collected as 1 mL fractions. Two bands were visible in the column and eluted separately. The presence of FITC conjugated CytC (CytC-FITC) in the first eluted band was confirmed by analyzing absorbance at 280 nm and 495 nm[46,51].

### Peptide assembly

The peptides were dissolved in hexafluoroisopropanol (HFIP) to remove any preformed aggregates and dried with nitrogen flow. The dried peptide samples were then dissolved in 40% acetonitrile/water, containing 0.1% of TFA and kept for getting assembled for a month[46,51].

### Redispersion

One millilitre of assembled peptide sample (2.5 mM) was taken in micro-centrifuge tube (1.5 mL) and was centrifuged at constant 4 °C temperature for 30 min at 10,000 rpm (rotor F45-30-11) in Eppendorf centrifuge 5804 R. The supernatant was removed to separate the pellet and this was followed with redispersion in the same volume of water.

### Sonication

To prepare short nanotubes, redispersed solution of mature nanotubes was sonicated using Biobase probe sonicator (φ2) for a different time frame. This resulted in mechanical force-induced scission, which was further shown in the size distribution bar diagram in Fig. 2d (calculated from TEM images, Supplementary Fig. 6, inset)[50].

### Amyloid-based nanomotors preparation

Briefly, 75 μL of free urease (100 μM stock in water) was mixed with 40 μL of the sonicated amyloid nanostructures (2.5 mM stock, redispersed in water) and 385 μL of water (maintaining Ac-KL concentration to be 200 μM). After one hour of incubation, the mixed solution was centrifuged at 14,000 rpm for 30 min at 4 °C and redispersed in water. The pellet containing urease-bound Ac-KL was washed twice by repeating the procedure of centrifugation and redispersion and the supernatants were collected for protein estimation with Bradford Reagent. For the motors containing both enzymes, apart from urease, CytC (20 μL from 500 μM stock in water) was additionally exposed to the nanotubes for binding.

### Protein estimation

Urease estimation studies for amyloid-based nanomotors were done with Bradford assay (Sigma Aldrich, following the protocol: http://www.sigmaaldrich.com/content/dam/sigmaaldrich/docs/Sigma/Bulletin/b6916bul.pdf).

After each centrifugation, supernatants were collected for the estimation of urease and CytC. Briefly, 1 mL of standard concentrations of BSA and unknown concentrations of enzyme in supernatants were mixed with Bradford reagent in separate test tubes and incubated for 15 min. Afterward, the absorbance was measured spectrophotometrically at λ = 595 nm. The unknown concentration was calculated from the standard plot. As a control, amyloid nanostructures without enzymes were subjected to the same process and the supernatants were exposed to Bradford Assay. This control showed negligible absorbance at 595 nm.

## Data availability

The authors declare that all the supporting data are provided in the main text and Supplementary Information. All raw data generated during the current study are available from the corresponding authors upon request. Source data are provided with this paper.

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

## Acknowledgements

D.D., P.G., and D.M. acknowledge Swarnajayanti (SB/SJF/2020-21/08), CRG SERB (CRG/2022/003607), Start-up Research Grant (SRG/2022/000043), and IISER Tirupati respectively. C.G., S.G., A.C., and P.B. acknowledge PMRF (0501088), INSPIRE, CSIR, and IISER Thiruvananthapuram respectively. We acknowledge Dr. Tapan Chandra Adhyapak for fruitful discussions.

## Author contributions

D.D. conceived the idea and supervised the overall project. C.G. and S.G. contributed equally to the work. C.G., S.G., and A.C. developed the idea, performed the experiments, and analyzed the data. P.B. and P.G. performed the molecular dynamics simulations and analyzed the data. D.M. contributed towards the development of the idea. All authors contributed to the discussion of results and writing the manuscript.

## Competing interests

The authors declare no competing interests.
