## [Peer Review File · Nature Communications]

Dual Enzyme-powered Chemotactic Cross β Amyloid based Functional NanomotorsReviewers' Comments:

Reviewer #1:

Remarks to the Author:

The manuscript reports on a new architecture for the fabrication of self-propelled nanomotors, based on amyloid peptide and powered by enzymes. The use of biological structures as building blocks of the chassis of these nanomotors is of high interest for future biomedical applications. These structures can be elongated as nanotubes, which is interesting for their dynamics but also challenging for tracking and for maintaining persistence lengths.

The work presents some noteworthy results on the use of new bio-structures for nanomotors and if this section is more comprehensively described, it will be of more significance to the field since it is novel and interesting for soft- and biocompatible nanomotors.

These nanomotors contain two types of enzymes, one for motion and another one for navigation. Why urease has been widely used in the literature and the claims are confirmed in this paper, the other enzymes are not used significantly, and this referee has some doubts on the real potential in navigation control, as claimed. Chemotaxis is a very exciting topic but difficult to justify in such small scales.

Authors used sonication to shorten the tubes and claim that sonicated tubes will move better than longer ones. However, there should be an optimization of the lengths for long persistence mobility. Similar observations with similar structure and the same enzyme was reported in X.Ma et al. J.Am.Chem.Soc. 2016 138 (42), 13782-13785 but not cited in this work. Authors could also study the directionality of such peptide-tubes depending on the length. Also, there is no asymmetry in this system as the enzymes are attached only on the outside. Can authors confirm this hypothesis? One of the great benefits of this peptide is that they should, in principle be able to precisely localize enzymes, but if I am not wrong, this has not been exploited in this work.

Do authors know the enzyme-dependant on the speed?

The main concern comes with the chemotaxis experiments:

1. Why urease nanomotors do not present chemotaxis as other researchers have reported, and there is a need of a second enzyme?
2. Why is this second enzyme moving towards the gradient? How is the enzyme distributed around the peptide tube?
3. A video demonstrating that the agarose gel-releasing chemicals is NOT inducing flows which produce drift on nanoparticles is required. In other words, control experiment using nanomotors (same structure) but with other enzymes and without enzymes.
4. Figure 4. Is not clear what the simulations in e, and g show. Why time difference between both panels?

The catalytic augmentation section is rather "off topic" and this referee does not understand the rationale of it in this manuscript. Could author elaborate more on that?

I find somehow that the conclusions do not describe the main claims of the paper and should be revised accordingly.

Reviewer #2:

Remarks to the Author:

In this work, Souvik Ghosh and coworkers describe nanotubes made of amyloid-forming peptides that are decorated by enzymes. The urease enzymes can render the tubes self-propelling. When a second enzyme is loaded, the self-propelling tubes can become directional in their motion. Specifically, CytC, a peroxidase is added that helps the nanotubes swim up a gradient.

Overall, the manuscript would be suitable for Nature Communications. However, the work on the chemotaxis is very superficially worked out. Please add a detailed description of the study of the particles. The chemotaxis should be tested with the same amount of detail as was done in Figure 3 for

the motility. Please add a confocal study where the directionality can be directly measured.

Specific points:

The introduction is vague. nanomotor chassis should be defined.

"futuristic applications" in the abstract. I'm not sure what that means other than applications in the future. Same for "exotic functions".

It is not clear from the introduction what this work contributes compared to other chemotactic soft matter motors.

The error bars in the manuscript should be defined

Reviewer #3:

Remarks to the Author:

In this work, the authors prepare β amyloid peptide-based nanomotors and investigate their dynamics. I find the work well-suited for a high-impact, general-interest journal such as Nature Communications. However, I think that the analysis of the trajectories ought to be improved. An analysis of the superdiffusivity of the nanomotors should be attempted since there are many possible types of it. The analysis should have the aim of clarifying what are the driving forces. Indeed, the authors also perform some simulations, but without the analysis of the trajectories there seem to be a gap between simulations and experiments.

Reviewer #1 (Remarks to the Author):

The manuscript reports on a new architecture for the fabrication of self-propelled nanomotors, based on amyloid peptide and powered by enzymes. The use of biological structures as building blocks of the chassis of these nanomotors is of high interest for future biomedical applications. These structures can be elongated as nanotubes, which is interesting for their dynamics but also challenging for tracking and for maintaining persistence lengths.

The work presents some noteworthy results on the use of new bio-structures for nanomotors and if this section is more comprehensively described, it will be of more significance to the field since it is novel and interesting for soft- and biocompatible nanomotors.

>> We appreciate the Reviewer's comment and assessment about the 'some noteworthy results.....'. We have now elaborated on the significance of such biocompatible systems in the introduction (page 2, Revised MS, yellow highlight).

These nanomotors contain two types of enzymes, one for motion and another one for navigation. Why urease has been widely used in the literature and the claims are confirmed in this paper, the other enzymes are not used significantly, and this referee has some doubts on the real potential in navigation control, as claimed. Chemotaxis is a very exciting topic but difficult to justify in such small scales.

>> As pointed out by the Reviewer, urease has been widely used for the motility of nanomotor (New additional ref: <https://pubs.acs.org/doi/10.1021/acs.nanolett.5b03100> and <https://doi.org/10.1021/acs.langmuir.9b03315>, and also existing ref: 49 and 54 in Revised MS). However, the use of the other enzyme, cytochrome C (CytC), a heme protein that promotes oxidoreduction (A. I. Bunea, et al., *Chem. Commun.* **2013**, 49, 8803-8805), is less common in the field of nanomotors. Importantly, CytC-loaded nanotubes were found to exhibit enhanced diffusion at various pyrogallol concentrations (Supplementary Fig. 20, Revised SI). The primary power source of amylobots is derived from urease while the navigational potential originated from the CytC as supported by experiments.

Authors used sonication to shorten the tubes and claim that sonicated tubes will move better than longer ones. However, there should be an optimization of the lengths for long persistence mobility. Similar observations with similar structure and the same enzyme was reported in X.Ma et al. *J.Am.Chem.Soc.* 2016 138 (42), 13782-13785 but not cited in this work. Authors could also study the directionality of such peptide-tubes depending on the length.

>> We thank the Reviewer for the lead (now cited as ref. 58, Revised MS). We have now used different sonication times (15 min, 30 min, 45 min) in an attempt to vary the length of the nanotubes. From TEM, size distribution histograms were derived and the average lengths of the nanotubes were calculated (for 15, 30, and 45 min of sonication, the average lengths were found to be 560.9 ± 27.7 nm, 266.5 ± 67.7 nm, and 155.6 ± 48.4 nm, respectively, for all the cases $N=30$, Supplementary Fig. 6, inset). The nanotubular morphologies were maintained throughout. Notably, an increase in speed with the decrease in the persistence length of the nanotubes was observed (Supplementary Fig. 6, Revised SI). Further, no change in directionality was observed with the tubes sonicated for different time scales. The discussion is now included in the Revised MS (on page 6, yellow highlight).

Also, there is no asymmetry in this system as the enzymes are attached only on the outside. Can authors confirm this hypothesis? One of the great benefits of this peptide is that they should, in principle be able to precisely localize enzymes, but if I am not wrong, this has not been exploited in this work.

>> The amyloid assemblies have strong binding capabilities to noncovalently attach diverse hydrophobic guest molecules, from hydrophobic small molecules to 2D materials (Ref: *Nat. Chem.* **2017**, *9*, 805-809; *Angew. Chem. Int. Ed.* **2021**, *60*, 202-207). This allows the enzymes to bind both inside and outside of the hollow nanotubes. However, due to the homogeneously exposed amines (lysine) and hydrophobic residues (leucine), the amyloid surface does not offer interactions to precisely localize enzymes. The random attachment of the enzymes on the amyloid surface possibly led to the generation of local asymmetric sites on nanotubes. To investigate this hypothesis of local inhomogeneity of enzyme distribution, AFM investigations of the enzyme-bound nanotubes were performed. Indeed, different patch-like globular structures of proteins were distributed over the nanotube surface suggesting local inhomogeneity of binding (Supplementary Fig. 11, Revised SI). Also, TEM suggests similar inhomogeneity on the surface of the nanotubes. Notably, inhomogeneous binding of enzymes on nanotube surfaces was earlier observed by our group via AFM studies (*Angew. Chem. Int. Ed.* **2022**, *61*, e20220154; *Angew. Chem. Int. Ed.* **2015**, *54*, 6492–6495). Such asymmetric patches of enzyme on nanomotors have been shown to facilitate self-propulsion (for instance, T. Patiño, et al., *J. Am. Chem. Soc.* **2018**, *140*, 7896–7903, cited as 57, Revised MS). The discussion has been added in the Revised MS (Page 8, yellow highlight).

Do authors know the enzyme-dependant on the speed?

>> To monitor the dependency of the speed of the nanotubes, peptide nanotubes were exposed to various enzyme concentrations (1 to 20 μ M, Supplementary Fig. 7a). The MSD values were calculated and speed was found to be increased with increasing the loading of the enzyme (Supplementary Fig. 7, Revised SI, [urea]=50 mM, mentioned in Revised MS, page 6, yellow highlight).

The main concern comes with the chemotaxis experiments:

1. Why urease nanomotors do not present chemotaxis as other researchers have reported, and there is a need of a second enzyme?

>> We agree that there are elegant nanomotor systems in the literature that report chemotaxis generated from urease. The experimental design presented in this work does not utilize this known chemotactic capability of urease (none of the experiments had any gradient of urea as in all cases homogenous concentration of urea was used). The current work instead utilizes the propulsion capabilities of the urease along with a second enzyme (CytC) which installs the chemotaxis by reacting with an orthogonal substrate. Interestingly, a similar strategy of utilizing orthogonal processes for propulsion and chemotaxis is also seen in cell migration where two independent but interrelated processes (motility and directionality) are used (S. H. Larsen et al., *Proc. Nat. Acad. Sci. USA.* **1974**, *71*, 1239-1243; R. M. Macnab et al., *Proc. Nat. Acad. Sci. USA* **1972**, *69*, 2509-2512, D. WU, *Cell Research*, **2005**, *15*, 52-56). Aligned with the phenomena, the present work also deals with two different but interrelated processes. While the primary power

source is derived from the urease in presence of urea as a fuel to trigger the motility, chemotactic ability emanated from the enhanced diffusive capabilities of CytC towards the pyrogallol gradient (external stimuli/chemical cue). To date, this is the first work to demonstrate dual enzyme-loaded nanomotors that exhibit two distinct transport behaviour resulting in chemotactic motility thus foreshadowing the traits of extant biological processes.

2. Why is this second enzyme moving towards the gradient? How is the enzyme distributed around the peptide tube?

>> CytC is a class of protein that promotes oxidation of pyrogallol to purpurogallin through activation of hydrogen peroxide by the prosthetic hemin group (G.I. Berglund et al., *Nature* **2002**, 417, 463–468; N. Kapil et al., *Angew. Chem. Int. Ed.* **2015**, 54, 6492–6495). Notably, CytC-loaded nanotubes were found to exhibit enhanced diffusion in the presence of pyrogallol (Supplementary Fig. 20, Revised SI). Further, CytC has been previously shown to enhance the diffusivity of nanostructures in presence of H₂O₂ (A. I. Bunea et al., *Chem. Commun.* **2013**, 67, 8803-8805).

To show the enzyme distribution around the peptide nanotubes we have now performed AFM and TEM study with urease and CytC-loaded nanotubes as mentioned above (Supplementary Fig. 11, Revised SI).

3. A video demonstrating that the agarose gel-releasing chemicals is NOT inducing flows which produce drift on nanoparticles is required. In other words, control experiment using nanomotors (same structure) but with other enzymes and without enzymes.

>> As suggested by the Reviewer, we have now performed the control experiment using the same nanomotor systems with an orthogonal enzyme (glucose oxidase, GOx) along with urease and monitored the population of the nanotubes in the similar experimental setup (Supplementary Fig. 18a, Revised SI). There was no distributional bias towards the pyrogallol side (point 3, Supplementary Fig. 18b, Revised SI). Nanotubes without any enzymes and nanotubes with only urease also did not show any distributional bias (Supplementary Fig. 13, Revised SI). The discussions are now included in the Revised MS (page 9, yellow highlight).

4. Figure 4. Is not clear what the simulations in e, and g show. Why time difference between both panels?

>> Here we have mainly performed three different simulations. The substrate was modelled as a diffusive field at the one end of the rectangular simulation box of size (400 × 200) μm². The circular inoculation of 200 nanotubes of varying lengths (0.2 – 0.6) μm, were placed at position $x = L_x \times 2/3$, $y = L_y \times 1/3$. The equation of motion of nanotubes mainly contained three important terms as f_u (motility force due to urease), f_c (motility force due to CytC) and ∇C (the gradient of pyrogallol concentration) with a condition $f_u \gg f_c$ (for details see Computational Study section, Revised SI). To understand the role of CytC, control simulations were performed by keeping all other parameters same except $f_c = 0$ (no CytC).

The time scale in figure 4g was 15s as within this time frame the urease-nanotubes distributed throughout the box with no population bias towards any side. For figure 4e, 100 s was required to show the directional bias.

The catalytic augmentation section is rather “off topic” and this referee does not understand the rationale of it in this manuscript. Could author elaborate more on that?

>> The progress of non-aqueous enzyme catalysis in recent years has inspired us to explore the chemotactic motility of the peptide-based nanomotors to overcome the inter-particle diffusion limitations in a biphasic environment. The rationale behind the study is centered on the improvement of interfacial catalysis. Motivated from the observation that amylobots could show chemotaxis to the higher substrate gradient, we envisioned that the presence of the chemical cue (pyrogallol) in the toluene medium could help the propulsion of the CytC-loaded motile nanomotors present in the aqueous phase towards the organic medium and favour interfacial biocatalysis of CytC. Indeed, the improvement in the peroxidase activity by the urease-CytC loaded and urea-powered motile amylobots underpinned the efficiency of non-aqueous enzymatic catalysis. We are hopeful that the catalytic rate enhancement of enzymes in organic solvents by the functionalized amylobots could be adaptable in the future for chemical industries that use enzymes in non-aqueous heterogeneous environments. We have included the corresponding text in the Revised MS (page 12, yellow highlight).

I find somehow that the conclusions do not describe the main claims of the paper and should be revised accordingly.

>> As suggested by the Reviewer, we have now revised the conclusion (Revised MS, yellow highlight).

Reviewer #2 (Remarks to the Author):

In this work, Souvik Ghosh and coworkers describe nanotubes made of amyloid-forming peptides that are decorated by enzymes. The urease enzymes can render the tubes self-propelling. When a second enzyme is loaded, the self-propelling tubes can become directional in their motion. Specifically, CytC, a peroxidase is added that helps the nanotubes swim up a gradient. Overall, the manuscript would be suitable for Nature Communications.

>>We thank the Reviewer for the encouragement and appreciate his enthusiasm for the publication.

However, the work on the chemotaxis is very superficially worked out. Please add a detailed description of the study of the particles. The chemotaxis should be tested with the same amount of detail as was done in Figure 3 for the motility. Please add a confocal study where the directionality can be directly measured.

>> As suggested by the reviewer, we have now elaborated the chemotaxis section in a much-detailed manner. For this purpose, trajectory profiles of the nanomotors at different pyrogallol concentration gradients were now analyzed (Supplementary Fig. 25, Revised SI). The corresponding text is now included in the Revised MS (page 11, yellow highlight).

Further, a confocal study has been performed to demonstrate the chemotactic motility of the peptide nanomotors. Briefly, in an experimental setup (adapted from Y. Hong, et al., *Phys. Rev.*

Lett. **2007**, *99*, 178103) different concentrations of pyrogallol (0 mM, 50 mM, 100 mM) loaded capillaries (sealed at one end) were suspended in the buffer solution containing enzyme-loaded nanomotors and H₂O₂ in a 35 mm glass-bottom petriplate (Figure 4c in Revised MS demonstrate the experimental setup). CytC and urease were loaded on the nanomotors along with RITC-tagged urease (9:1 molar ratio). The population of nanomotors was probed at the opening of the capillaries under a confocal microscope. Notably, after ca. 15 minutes, the population at the capillary opening was found to be increased for higher concentrations of pyrogallol (Supplementary Fig. 24, Revised SI). In the absence of pyrogallol (with only buffer), no specific localization of the nanomotors near the capillary opening was observed. These results strongly indicated that the presence of pyrogallol (chemical cue) is critical for the directional motility of the dual enzyme-loaded nanotubes. The experimental observations and discussions are now included in the Revised SI (page 11, yellow highlight).

Specific points:

The introduction is vague. nanomotor chassis should be defined.

>> The introduction is now more focused and we have tried to be more specific in choosing the relevant words. Nanomotor chassis is defined now in the revised introduction (page 2, yellow highlight).

"futuristic applications" in the abstract. I'm not sure what that means other than applications in the future. Same for "exotic functions".

>> These terms have been corrected.

It is not clear from the introduction what this work contributes compared to other chemotactic soft matter motors.

>> We have now re-focused the introduction (Revised MS, page 2, yellow highlight).

The error bars in the manuscript should be defined

>> The errors were measured from the experiments performed in triplicates (three separate experiments) and are now defined in the Revised MS. For calculating the MSD values of 20 particles were tracked. This is now mentioned in the Revised MS.

Reviewer #3 (Remarks to the Author):

In this work, the authors prepare β amyloid peptide-based nanomotors and investigate their dynamics. I find the work well-suited for a high-impact, general-interest journal such as Nature Communications.

>> We thank the Reviewer's comment and appreciate the enthusiasm for its publication.

However, I think that the analysis of the trajectories ought to be improved. An analysis of the superdiffusivity of the nanomotors should be attempted since there are many possible types of it. The analysis should have the aim of clarifying what are the driving forces. Indeed, the authors

also perform some simulations, but without the analysis of the trajectories there seem to be a gap between simulations and experiments.

>> We have noted the Reviewers suggestion and performed a detailed analysis of the MSD trajectories (both experimental and simulated). To get better insight into the spatiotemporal dynamics of amylobots, we have fitted the MSD values as $MSD = 4D\Delta t^\alpha$, where Δt is the time interval, D is the diffusion constant and α is the MSD exponent. Depending upon MSD exponents, we characterized the type of diffusion i.e. how fast or slow the particles are diffusing in the space (Y. Zhang, et al., *Nat. Rev. Chem.* **2021**, *5*, 500–510; S. Song, et al., *J. Am. Chem. Soc.* **2022**, *144*, 30, 13831–13838). $\alpha = 1$ indicates the Brownian diffusion, $\alpha < 1$ implies the sub-diffusion, and $\alpha > 1$ specifies the superdiffusion. Supplementary Fig. 10 demonstrates the fitting of experimental and simulated MSD plots as a function of time intervals respectively. From these two plots, it becomes clear that in absence of urea (motility force) the nanotubes are showing Brownian diffusion ($\alpha \sim 1$). However, for higher values of urea concentration (motility force), the amylobots are showing superdiffusive motion ($\alpha > 1$) suggesting urea concentration (motility) driven fast dispersal. This discussion is now included in the revised MS (page 8, yellow highlight).

Reviewers' Comments:

Reviewer #1:

Remarks to the Author:

The authors did not convincingly explained one common concern from the referees: the chemotactic phenomena.

This still remains a major problem in this paper. For example, if there is no gradient in the solution (are responded by the authors), what is represented (blue and yellow gradients) in the simulations and if there is not, where is the asymmetrical distribution of products originates in order to provide a directionality of the rods?

If one pays attention to the supporting videos on the tracking of motion, it is clear that many other rods were not tracked. It would be more significant to track different on the same ROI to see the real dispersion in the motion, which seems much larger than that plotted.

Mechanism of motion: if the authors are so sure about the novelty and uniqueness of the orthogonal reactions with dual enzymes, this should be a major part of the paper. However, it is not really highlighted nor deeply explained. The community would benefit from a thorough mechanistic explanation.

Thus, for the above comments, I feel that unfortunately this manuscript is not in conditions to be published in Nature Comm.

Reviewer #2:

Remarks to the Author:

The authors have addressed all my comments, and I now support accepting the work. I would strongly advise though to change the 3D graph in figure 5D to a 2D graph. You lose information like this. Specifically, the data point at 0 mM range from 2-5 $\mu\text{M}/\text{min}$ due to the 3D nature of the bar. In a 2D plot, it's immediately clear what you mean. Nevertheless, I leave it up to the authors and do not have to re-review te work.

Reviewer #3:

Remarks to the Author:

In my opionion, the revised version is suitable for publication in Nature Comms.

Reviewer #1 (Remarks to the Author):

The authors did not convincingly explained one common concern from the referees: the chemotactic phenomena. This still remains a major problem in this paper. For example, if there is no gradient in the solution (are responded by the authors), what is represented (blue and yellow gradients) in the simulations and if there is not, where is the asymmetrical distribution of products originates in order to provide a directionality of the rods?

>> There is a gradient of pyrogallol only in the medium (as depicted by yellow colour diffusion to the blue background in Fig. 4 e) which acts as a chemical cue required for the chemotaxis by the CytC loaded nanomotors. The nanomotors were also loaded with urease for motility. Hence, the tubes gradually move towards the pyrogallol gradient as the CytC reacts with its substrate pyrogallol while the propulsion is due to the urease. The directionality is purely due to chemical gradient.

If one pays attention to the supporting videos on the tracking of motion, it is clear that many other rods were not tracked. It would be more significant to track different on the same ROI to see the real dispersion in the motion, which seems much larger than that plotted.

>> Additional nanorods on the same ROI were tracked (for 50 and 100 mM urea concentration, Supplementary Fig. 6, Revised SI). These new data points are in the similar range as the previously observed MSD.

Mechanism of motion: if the authors are so sure about the novelty and uniqueness of the orthogonal reactions with dual enzymes, this should be a major part of the paper. However, it is not really highlighted nor deeply explained. The community would benefit from a thorough mechanistic explanation. Thus, for the above comments, I feel that unfortunately this manuscript is not in conditions to be published in Nature Comm.

>> As suggested, we have now focused more on the role of dual enzymes in showing chemotactic motility by orthogonal reactions. Additional discussions are now included on pages 2, 9, and 13 in the Revised MS (yellow highlights).

Reviewer #2 (Remarks to the Author):

The authors have addressed all my comments, and I now support accepting the work. I would strongly advise though to change the 3D graph in figure 5D to a 2D graph. You lose information like this. Specifically, the data point at 0 mM range from 2-5 $\mu\text{M}/\text{min}$ due to the 3D nature of the bar. In a 2D plot, it's immediately clear what you mean. Nevertheless, I leave it up to the authors and do not have to re-review te work.

>>We thank the Reviewer for the enthusiasm for publication. We have now modified the 3D bar to a 2D bar in figure 5b.

Reviewer #3 (Remarks to the Author):

In my opinion, the revised version is suitable for publication in Nature Comms.

>>We thank the Reviewer's comment and appreciate the enthusiasm for its publication.

Reviewers' Comments:

Reviewer #1:

Remarks to the Author:

Dear all. I am sorry but I still have serious fundamental concerns on the principles claimed in this work and I am not satisfied with the answers and I feel that authors do not clearly understand the mechanism of motion and directionality of the rods.

1. There is no urea gradient, now it is clear from the reply. Nanomotors move because of urease reaction with urea substrate. Then, what is the contribution on the motion from the CytC reaction with pyrogallol? To me, chemotaxis should not take place if CytC is not playing a role in the motion. What if authors remove urea and keep pyrogallol gradient? That is a key experiment.

2. Although authors add a S.I figure 6 with different tracking trajectories, I do not find the videos tracked where I would like to see many particles moving at the same time in the same ROI. That was my question. I do not know the origin of those tracking trajectories.

3.Regarding the mechanism of motion, authors replied that further discussion has been added in page 2, 9, and 13. To me, these "discussions" are not sufficient at all. Authors basically repeat what they claimed before. When I mean a mechanistic explanation I really mean that there is no physics in the manuscript explaining the rational why having two enzymes (orthogonal reactions) would lead to a directional motion whatsoever. Maybe I am not understanding the manuscript, but I wonder if other readers would understand why the chemotaxis is taking place. What is the connection between the two enzymes which make them "SENSE" higher substrate gradient? (page 2). What is the "sensing mechanism" claimed here?

From authors:

Page 2:

More specifically, while motility is regulated by the consumption of energy, directionality is controlled by sensing of higher substrate gradient by a spatial sensing mechanism

Page 9

These results indicated that chemotaxis originated from the orthogonal enzyme CytC loaded on the nanotubes (along with urease) when a spatial gradient of pyrogallol was present in the medium.

Page 13

Importantly, this two-enzyme-regulated chemotactic movement shown by the synthetically accessed soft nanoconstructs further signifies the importance of the distribution of the payload observed in the contemporary motile biological systems.

Reviewer #4:

Remarks to the Author:

Please see the attached pdf.

Overall evaluation: The presentation and explanation of nanotube chemotaxis is flawed. I recommend the authors carefully respond to my comments on chemotaxis, and re-examine their results, before this manuscript can be considered for publication.

My evaluation is mostly focused on the chemotaxis of the reported enzyme-functionalized nanotubes, since this seems to be the focal point of reviewer 3.

-Scientific novelty

It is not unheard of that an enzyme-functionalized colloidal particle can show chemotaxis. A seminal study on this subject is from Prof. Ayusman Sen:

Somasundar, A.; Ghosh, S.; Mohajerani, F.; Massenburg, L. N.; Yang, T.; Cremer, P. S.; Velegol, D.; Sen, A. Positive and Negative Chemotaxis of Enzyme-Coated Liposome Motors. *Nature Nanotechnology* 2019, 14 (12), 1129–1134.

And this topic has been reviewed a few times over the past few years. For example:

- Feng, Mudong, and Michael K. Gilson. "Enhanced diffusion and chemotaxis of enzymes." *Annual Review of Biophysics* 49 (2020): 87-105.
- Gao, Chao, Ye Feng, Daniela A. Wilson, Yingfeng Tu, and Fei Peng. "Micro-Nano Motors with Taxis Behavior: Principles, Designs, and Biomedical Applications." *Small* 18, no. 15 (2022): 2106263.
- Liebchen, Benno, and Hartmut Löwen. "Synthetic chemotaxis and collective behavior in active matter." *Accounts of chemical research* 51, no. 12 (2018): 2982-2990.

Although the observation is not entirely ground-breaking, I still think there is enough scientific novelty reported here that merits its publication, provided that the chemotaxis reported here is true, and reasonably explained (see below).

-mechanism of chemotaxis

The biggest concern I have, somewhat aligned with reviewer 3, is that this manuscript lacks a proper discussion on mechanisms. However, by this I don't mean the mechanism for self-propulsion, which is well-reported for enzyme-coated colloids in its substrate. For example, a urease-coated motor moves in urea most likely because of ionic self-diffusiophoresis, which originates from the diffusivity differences between the ions released. (The authors can elaborate on this mechanism a bit if they haven't already done so.)

In fact, I think Reviewer 3 was confusing the self-propulsion of a nanotube, which is powered mostly by urease as the authors have clearly stated, with its chemotaxis, which is enabled by a second enzyme cytc as the authors have also clearly stated. A gradient of urea is not needed for its propulsion (uniform fuel is enough for power), or its chemotaxis (claimed to be due to cytc, not urease). I would say there is no synergy between these two enzymes. One is the "leg", and the other is the "nose".

On the other hand, I am mostly concerned with the mechanism of chemotaxis, and whether it truly happened in the experiments reported here.

First, I'm not entirely convinced that the directional migration of a nanotube population is chemotaxis. There are certainly other possibilities, with convection being the number

I suspect. Convection can originate from a few sources, such as uneven substrate, wind, or temperature gradient. More subtly, it can also come from a density gradient caused by the addition of chemicals, or osmotic flows that originate from a chemical gradient. All of the above can lead to collective migration of nanotubes, yet none is chemotaxis.

To eliminate these possibilities, the authors have performed a few control experiments, which are very applaudable and indeed critical. However, I am having a very hard time understanding how these experiments were performed, and exactly which possibility they are trying to eliminate. Very little was said about the reasoning behind any of the experiment design, or what the result of any experiment means.

For example: What is on the nanotube for Fig. 15? What is “free cytc” in Fig. 16 and 17? What does the result of Fig. 18 mean? How does Fig. 19 prove there is no drift? What is the grey P dot in Fig. 19? Was any pyrogallol added in Fig. 19?

Too many questions, yet too little clue.

To solve the above issue, I strongly recommend the authors greatly expand the following sentence into one full paragraph: “*Notably, different control systems with varying conditions were unable to display the chemotactic migration of the nanomotors (Supplementary Figs. 14-18). The possibility of any drift on nanotubes that could have occurred by the chemical cue released from the agarose gel was ruled out by performing control experiments with the orthogonal enzyme (Supplementary Fig. 19, see SI for details).*”. In this new paragraph, please clearly state the various possibilities that could cause migration beyond chemotaxis. Then, please explain the strategies to eliminate these possibilities, followed by clear explanations of how experimental results have done just that.

One key experiment that is critical to rule out other possibilities, yet missing from the current study, is to mix in the same experiment “active” nanotubes (i.e. one that is supposed to show chemotaxis) with passive ones (i.e. those that do not chemotaxis), and see if they migrate differently. Because most of the suspected possibilities I’ve listed earlier do not distinguish between active or passive nanotubes, this experiment will be very powerful in confirming that nanotubes are indeed showing chemotaxis.

Finally, I’d like to see a reasonably adequate discussion on why these nanotubes chemotaxis (assuming they do). This question is the same as the one from reviewer 3: “What is the “sensing mechanism” claimed here?” This is not trivial. Over the years there has been some controversy about whether rod shapes nanomotors show positive chemotaxis, first reported by Hong et al. (Phys. Rev. Lett. 2007, 99 (17), 582–584). This kind of chemotaxis is quite counter-intuitive, because one would naively think that a motor would over time migrate away from regions where they move faster, and be collected at regions where they move more slowly. Why do the authors think their nanotubes should be collected at high concentrations of substrates? Do their experiments give any fundamental insight of how this happens? If so, this would be a quite substantial achievement to the entire field of active matter.

Without this mechanistic understanding, it is difficult to answer the following question: why doesn’t a urease-functionalized nanotube chemotaxis in a urease gradient, which according to the author is much more powerful?

Minor issues:

- The phrase “nanotube-bound urease” is misleading, and should be changed to “urease-bound nanotube”.

-I don't understand what this result means: "*Notably, rapid generation of purpurogallin was observed in the gel buffer interface with a 7-fold higher rate than the controls without urea (Supplementary Figs. 20, 23)*"

-The description of the following experiment is misleading “*Different concentrations of pyrogallol (0 mM, 50 mM, 100 mM) loaded glass capillaries...*”. The main text only mentions “enzyme-loaded nanomotors and H₂O₂”, which to me suggested that the motor was functionalized solely with ctyc and no urease or urea was present in this experiment. This would be an interesting experiment to show that ctyc alone could cause chemotaxis. However, I later found in Supporting Information that “Dual enzyme-loaded nanomotors along with urea were added to the buffer system”. This is misleading to say the least. With this new knowledge, I can't help but wonder what is the point of performing this set of experiment, because it is basically a repetition of the results in Fig. 4.

Reviewer #1 (Remarks to the Author):

Dear all. I am sorry but I still have serious fundamental concerns on the principles claimed in this work and I am not satisfied with the answers and I feel that authors do not clearly understand the mechanism of motion and directionality of the rods.

>> We thank the reviewer for the critical assessment. To address the concerns, we have conducted additional experiments (also some suggested by reviewer 4) and proposed a mechanism based on the results.

1. There is no urea gradient, now it is clear from the reply. Nanomotors move because of urease reaction with urea substrate. Then, what is the contribution on the motion from the CytC reaction with pyrogallol? To me, chemotaxis should not take place if CytC is not playing a role in the motion. What if authors remove urea and keep pyrogallol gradient? That is a key experiment.

>> The main experiment was always done in the presence of an equilibrated concentration of urea, i.e., in the absence of any urea gradient. Urea gradient was never used in any of the previous submissions. Also, from the very initial submission, the control experiment was included where urea was absent while the pyrogallol gradient was present (Supplementary Fig. 22b). Unfortunately, the reviewer had missed these experiments in our previous submissions.

For the control without urea, we have now monitored for longer time periods (ca. 1.5h instead of 40 min, CLSM study with capillary filled with pyrogallol, see Supplementary Fig. 27c, Revised SI, yellow highlights). We did observe a modest increase in the overall population of the nanomotors near the capillary opening. It would be important to mention here that in the presence of urea, only after ca. 15-20 min localization of the urease-CytC motors could be observed. In combination, these results suggest that motors in the absence of urea do have a subdued yet finite chemotactic propensity towards a gradient of pyrogallol, and it takes a longer time to achieve this.

Now, regarding the central question of the origin of chemotaxis in the CytC-urease-loaded nanotubes in a gradient of pyrogallol, we have proposed the following mechanism. For motility, urease uses chemo-mechanical energy conversion via the generation of a local electric field from the difference in diffusivities of the oppositely charged ionic species (as shown by many reports, *J. Am. Chem. Soc.* **2013**, *135*, 1406–1414, *J. Am. Chem. Soc.* **2016**, *138*, 13782–13785). Hence, when urea was uniformly distributed in the medium (and no CytC + Pyrogallol gradient), the urease-bound nanotubes only showed the persistent Brownian motion without any specific bias towards any direction. The persistent Brownian motion indicated spatial asymmetry in the urease reaction, which was presumably due to the inhomogeneous binding of the enzymes on the rod-like nanomotor chassis (*J. Am. Chem. Soc.* **2018**, *140*, 7896–7903, inhomogeneous binding could also be seen from AFM images, Supplementary Fig. 12). However, the persistent motion of the nanomotors did not create any directionality at long durations due to the rotational Brownian motion (*PLoS Biol.*, **2016**, *14*, e1002463). Further, any reaction fluctuations may also contribute to rotational motion due to transient asymmetries (*PLoS Biol.*, **2016**, *14*, e1002463). In short, the nanomotors had persistent motion without any specific bias in the directionality as the substrate urea was homogeneously distributed.

Now, to rationalize the chemotaxis, we propose that an additional mechanism leads to a decline in this random rotational Brownian motion of the nanomotors. In this context, it is known that

the chemotaxis of Janus particles involves their active rotation, coupled with the direction of fuel gradient (*Nano Lett.* **2018**, *18*, 5345–5349; *Biochemistry* **2018**, *57*, 6256–6263). A detailed theoretical derivation of this active rotational component can also be found in the literature (*Physics of Fluids* **2021**, *33*, 032011). We propose a similar mechanism in our system in the inhomogeneous presence of an additional weak enzyme (CytC). The random rotation of the nanomotor chassis presumably diminishes due to the CytC-induced active rotations towards the pyrogallol gradient despite the weak catalytic proficiency. This installs a weak external field which eventually restricts the random rotations of the urease+CytC-powered self-diffusiophoretic motors biasing the direction of motion towards the higher concentration region of pyrogallol. In other words, this additional interaction (CytC-pyrogallol) disrupts the overall randomness in directionality in the strong persistent motion powered by urease. Subsequently, the motors now make a gradual choice towards the external bias of the gradient of the CytC-substrate (*PLoS Biol.*, **2016**, *14*, e1002463, *EPL* **2013**, *103*, 60009). However, after rotation, there would still be an equal possibility of the motors to go against or towards the gradient. We argue that the choice towards the pyrogallol gradient could partially be attributed to the weak yet finite diffusivity of the CytC (CytC-loaded nanotubes in the absence of urease indeed showed enhanced diffusivity (at 100 mM pyrogallol), although the diffusivity was significantly lesser compared to urease-CytC loaded nanotubes, $D_{urease/cytc} = 6.99 \pm 0.05 \mu\text{m}^2/\text{s}$, $D_{cytc} = 0.42 \pm 0.02 \mu\text{m}^2/\text{s}$ in buffer). However, understanding the interplay between the orthogonal enzymes, that breaks the symmetry of equal possibility of moving against or towards the pyrogallol gradient, requires further dedicated investigations, and our current efforts are towards this front.

2. Although authors add a S.I figure 6 with different tracking trajectories, I do not find the videos tracked where I would like to see many particles moving at the same time in the same ROI. That was my question. I do not know the origin of those tracking trajectories.

>> As suggested by the Reviewer, particles have been tracked freshly and the videos are provided in the supplementary information (Supplementary Videos 3 and 4, Revised SI).

3. Regarding the mechanism of motion, authors replied that further discussion has been added in page 2, 9, and 13. To me, these “discussions” are not sufficient at all. Authors basically repeat what they claimed before. When I mean a mechanistic explanation I really mean that there is no physics in the manuscript explaining the rational why having two enzymes (orthogonal reactions) would lead to a directional motion whatsoever. Maybe I am not understanding the manuscript, but I wonder if other readers would understand why the chemotaxis is taking place. What is the connection between the two enzymes which make them “SENSE” higher substrate gradient? (page 2). What is the “sensing mechanism” claimed here?

From authors:

Page 2: More specifically, while motility is regulated by the consumption of energy, directionality is controlled by sensing of higher substrate gradient by a spatial sensing mechanism

Page 9

These results indicated that chemotaxis originated from the orthogonal enzyme CytC loaded on the nanotubes (along with urease) when a spatial gradient of pyrogallol was present in the medium.

Page 13

Importantly, this two-enzyme-regulated chemotactic movement shown by the synthetically accessed soft nanoconstructs further signifies the importance of the distribution of the payload observed in the contemporary motile biological systems.

>> The proposed mechanism is described in the above point. We have now included the mechanism as a separate section: Proposed mechanism of chemotactic motility (Pages 14-15, Revised MS).

Reviewer #4 (Remarks to the Author):

Overall evaluation: The presentation and explanation of nanotube chemotaxis is flawed. I recommend the authors carefully respond to my comments on chemotaxis, and re-examine their results, before this manuscript can be considered for publication. My evaluation is mostly focused on the chemotaxis of the reported enzyme-functionalized nanotubes, since this seems to be the focal point of reviewer 3.

-Scientific novelty

It is not unheard of that an enzyme-functionalized colloidal particle can show chemotaxis.

A seminal study on this subject is from Prof. Ayusman Sen: Somasundar, A.; Ghosh, S.; Mohajerani, F.; Massenbarg, L. N.; Yang, T.; Cremer, P. S.; Velegol, D.;

Sen, A. Positive and Negative Chemotaxis of Enzyme-Coated Liposome Motors. *Nature Nanotechnology* 2019, 14 (12), 1129–1134

And this topic has been reviewed a few times over the past few years. For example:

- Feng, Mudong, and Michael K. Gilson. "Enhanced diffusion and chemotaxis of enzymes." *Annual Review of Biophysics* 49 (2020): 87-105.

- Gao, Chao, Ye Feng, Daniela A. Wilson, Yingfeng Tu, and Fei Peng. "Micro-Nano Motors with Taxis Behavior: Principles, Designs, and Biomedical Applications." *Small* 18, no. 15 (2022): 2106263.

- Liebchen, Benno, and Hartmut Löwen. "Synthetic chemotaxis and collective behavior in active matter." *Accounts of chemical research* 51, no. 12 (2018): 2982- 2990.

>> We appreciate the careful review of the reviewer. We have now provided robust experiments for chemotaxis (see below). Also, we have included a proposed mechanism. We appreciate the reference leads provided by the reviewer.

Although the observation is not entirely ground-breaking, I still think there is enough scientific novelty reported here that merits its publication, provided that the chemotaxis reported here is true, and reasonably explained (see below).

>> We appreciate the comment about scientific novelty, and we thank the Reviewer for the enthusiasm for publication. However, we are concerned about the criticisms. We have conducted additional experiments (suggested by both reviewers) and carefully modified the manuscript.

-mechanism of chemotaxis

The biggest concern I have, somewhat aligned with reviewer 3, is that this manuscript lacks a proper discussion on mechanisms. However, by this I don't mean the mechanism for self-

propulsion, which is well-reported for enzyme-coated colloids in its substrate. For example, a urease-coated motor moves in urea most likely because of ionic self-diffusiophoresis, which originates from the diffusivity differences between the ions released. (The authors can elaborate on this mechanism a bit if they haven't already done so.)

In fact, I think Reviewer 3 was confusing the self-propulsion of a nanotube, which is powered mostly by urease as the authors have clearly stated, with its chemotaxis, which is enabled by a second enzyme cytc as the authors have also clearly stated. A gradient of urea is not needed for its propulsion (uniform fuel is enough for power), or its chemotaxis (claimed to be due to cytc, not urease). I would say there is no synergy between these two enzymes. One is the "leg", and the other is the "nose".

>> As suggested by the Reviewer, the discussion on the self-diffusiophoresis has been elaborated in the revised MS (page 5, yellow highlight). The experiment with active and passive motors has also been included now, please see next point.

On the other hand, I am mostly concerned with the mechanism of chemotaxis, and whether it truly happened in the experiments reported here.

First, I'm not entirely convinced that the directional migration of a nanotube population is chemotaxis. There are certainly other possibilities, with convection being the number 1 suspect. Convection can originate from a few sources, such as uneven substrate, wind, or temperature gradient. More subtly, it can also come from a density gradient caused by the addition of chemicals, or osmotic flows that originate from a chemical gradient. All of the above can lead to collective migration of nanotubes, yet none is chemotaxis.

>> We understand the Reviewer's concern.

First, to eliminate the possibilities of convective flow generated from different sources, all possible measures were taken, many controls were done that are listed below, and all the experiments were done multiple times (minimum of three times).

To prevent air-induced convection the petri plate was covered while performing the experiments. Further, all the experiments were performed at room temperature (temperature variation of less than ± 1 °C). The temperature variation in the local aqueous environment for the catalytic reactions by urea-urease and CytC-pyrogallol is expected to be negligible in aqueous medium (although not measured due to lack of proper setup). The possibility of any convection on nanotubes that could have occurred by the density gradient caused by the addition of chemical (pyrogallol gradient) was ruled out by performing control experiments with another enzyme (GOx, Supplementary Fig. 19, see SI for details). Instead of CytC, GOx was loaded along with urease. However, no directional bias was observed towards the pyrogallol gradient ('P' stands for pyrogallol in the reservoir, see Supplementary Fig. 19). This suggested that the uneven substrate density gradient caused by chemicals or probable osmotic flows originating from pyrogallol gradient did not induce any collective migration.

To eliminate these possibilities, the authors have performed a few control experiments, which are very applaudable and indeed critical. However, I am having a very hard time understanding how these experiments were performed, and exactly which possibility they are trying to eliminate. Very little was said about the reasoning behind any of the experiment design, or what the result of any experiment means.

For example: What is on the nanotube for Fig. 15? What is “free cytc” in Fig. 16 and 17? What does the result of Fig. 18 mean? How does Fig. 19 prove there is no drift? What is the grey P dot in Fig. 19? Was any pyrogallol added in Fig. 19?

Too many questions, yet too little clue.

To solve the above issue, I strongly recommend the authors greatly expand the following sentence into one full paragraph: “Notably, different control systems with varying conditions were unable to display the chemotactic migration of the nanomotors (Supplementary Figs. 14-18). The possibility of any drift on nanotubes that could have occurred by the chemical cue released from the agarose gel was ruled out by performing control experiments with the orthogonal enzyme (Supplementary Fig. 19, see SI for details).”. In this new paragraph, please clearly state the various possibilities that could cause migration beyond chemotaxis. Then, please explain the strategies to eliminate these possibilities, followed by clear explanations of how experimental results have done just that.

>> As recommended by the Reviewer, we have now provided a detailed explanation for each of the control experiments (pages 9-10, yellow highlight, Revised MS). The demonstrated control experiments clearly eliminate different possibilities that could lead to the enhancement in the population of the nanotubes to the higher substrate concentration of pyrogallol. The paragraph is added below and also added on pages 9-10 (yellow highlight) of the main text (Revised MS).

Firstly, to show the combined effect of the CytC and the pyrogallol gradient, control experiments were performed excluding one of the components at a time from the system. When the experiment was performed with only urease-loaded nanotubes (CytC was absent but pyrogallol gradient was present), a random trend in the change of population of the nanotubes was observed (Supplementary Fig. 14). Also, in the absence of a pyrogallol gradient (the reservoir ‘B’ was filled with buffer instead of pyrogallol), a similar observation with no distributional bias was seen (Supplementary Fig. 15). Both results strongly suggest the role of the additional enzyme (CytC) for the observed chemotactic migration. In addition, freely diffusing CytC was added in the medium along with only urease-loaded nanotubes to probe the possibility of whether the non-specific binding of the product generated from the catalytic activity could show the directionality of the nanotubes towards the pyrogallol gradient. However, no specific increment in the population of the nanotubes was observed (Supplementary Fig. 16). Further, the agarose gel was allowed to release the pyrogallol for a longer time period of time (ca. 8h) which would lead to equilibrated pyrogallol concentrations with a homogenous distribution. In this case, as well, the population of nanotubes at points 3 and point 1 failed to show any particular bias (Supplementary Fig. 17). To rule out any unexpected effects from the combination of reactions (peroxidase by CytC with pyrogallol and the urea-urease reaction), the gel was mixed with free CytC and pyrogallol was added at the reservoir and the system was equilibrated (ca. 1h) before the addition of the nanomotors (only urease loaded nanotubes) in the medium (at point 2). Again, no distribution bias in the population could be observed eliminating any cumulative effects generated from the orthogonal reactions (Supplementary Fig. 18). To eliminate the possibilities of convective flow generated from different sources, all possible measures were taken, several controls were done that are listed below, and all the experiments were done multiple times. To prevent air-induced convection the petri plate was covered while performing the experiments. Further, all the experiments were performed at room temperature (temperature variation of less than $\pm 1^\circ\text{C}$). The temperature variation in the local aqueous environment for the catalytic

reactions by urea-urease and CytC-pyrogallol is expected to be negligible. The possibility of any drift flow on nanotubes that could have occurred by the chemical cue released from the agarose gel was ruled out by performing control experiments with another enzyme (GOx, Supplementary Fig. 19, see SI for details). Instead of CytC, in this case, GOx was loaded along with urease. However, no directional bias was observed towards the pyrogallol gradient (Supplementary Fig. 19, 'P' stands for pyrogallol in the reservoir). This suggested that uneven substrate distribution, density gradient caused by chemicals, or probable osmotic flows originating from pyrogallol gradient did not induce any collective migration.

One key experiment that is critical to rule out other possibilities, yet missing from the current study, is to mix in the same experiment "active" nanotubes (i.e. one that is supposed to show chemotaxis) with passive ones (i.e. those that do not chemotaxis), and see if they migrate differently. Because most of the suspected possibilities I've listed earlier do not distinguish between active or passive nanotubes, this experiment will be very powerful in confirming that nanotubes are indeed showing chemotaxis.

>> We thank the Reviewer for suggesting this key experiment.

As suggested by the Reviewer, we prepared both active (urease and CytC loaded) and passive (only urease loaded) nanotubes and monitored their migration in the presence of homogenous urea conditions and a pyrogallol gradient. For the experiment, we have taken a similar setup as used to show chemotaxis (as used in Fig. 4c, Revised MS). Indeed, the active nanotubes showed a propensity towards the higher substrate gradient, whereas the passive ones were mostly dispersed in the medium. Briefly, a capillary filled with pyrogallol (sealed at one end) was suspended in a 35 mm glass bottom petri plate, and a mixed system of active and passive nanomotors was dispersed in the medium. For differentiating the systems, active nanotubes were loaded with FITC-tagged urease (loaded along with untagged urease in 1:9 molar ratio), while the passive nanotubes were loaded with RITC-tagged urease, respectively. The fluorescence intensity (FI) was monitored at the capillary opening using the confocal microscope (experiments were performed in triplicates, and the fluorescence intensity for all the experiments was normalized to 1). The normalized fluorescence intensity of FITC (active ones) was found to be gradually increasing with time at the opening of the capillary aperture, whereas a random distribution was suggested from the normalized FI profile for RITC (passive ones, Supplementary Fig. 28, Revised SI). Thus, the enhancement in the FI (for FITC) suggested the increase of the population of the active nanotubes towards the higher pyrogallol gradient, while the passive nanotubes were mostly randomly dispersed in the medium. This observation clearly indicated the role of CytC (present on the nanotubes along with urease) in the navigation of the motors towards the higher pyrogallol gradient.

Finally, I'd like to see a reasonably adequate discussion on why these nanotubes chemotaxis (assuming they do). This question is the same as the one from reviewer 3: "What is the "sensing mechanism" claimed here?" This is not trivial. Over the years there has been some controversy about whether rod shapes nanomotors show positive chemotaxis, first reported by Hong et al. (Phys. Rev. Lett. 2007, 99 (17), 582–584). This kind of chemotaxis is quite counter-intuitive, because one would naively think that a motor would over time migrate away from regions where they move faster, and be collected at regions where they move more slowly. Why do the authors

think their nanotubes should be collected at high concentrations of substrates? Do their experiments give any fundamental insight of how this happens? If so, this would be a quite substantial achievement to the entire field of active matter.

Without this mechanistic understanding, it is difficult to answer the following question: why doesn't a urease-functionalized nanotube chemotaxis in a urease gradient, which according to the author is much more powerful?

>> Regarding the central question of the origin of chemotaxis in the CytC-urease-loaded nanotubes in a gradient of pyrogallol, we have proposed the following mechanism. When urea was uniformly distributed in the medium (without CytC + Pyrogallol gradient), the urease-bound nanotubes only showed persistent Brownian motion without any specific bias towards any direction. The persistent Brownian motion indicated spatial asymmetry in the urease reaction, which was presumably due to the inhomogeneous binding of the enzymes on the rod-like nanomotor chassis (*J. Am. Chem. Soc.* **2018**, *140*, 7896–7903, inhomogeneous binding could also be seen from AFM images, Supplementary Fig. 12). However, the persistent motion of the nanomotors did not create any directionality at long durations due to the rotational Brownian motion (*PLoS Biol.*, **2016**, *14*, e1002463). Further, any reaction fluctuations may also contribute to rotational motion due to transient asymmetry (*PLoS Biol.*, **2016**, *14*, e1002463). Thus, the nanomotors had persistent motion without any specific bias in the directionality of the overall motion as the substrate urea was homogeneously distributed.

To rationalize the chemotaxis, we propose that an additional mechanism leads to a decline in this random rotational Brownian motion of the nanomotors. In this context, it is known that the chemotaxis of Janus particles involves their active rotation coupled with the direction of fuel gradient (*Nano Lett.* **2018**, *18*, 5345–5349). A detailed theoretical derivation of this active rotational component can also be found in the literature (*Physics of Fluids* **2021**, *33*, 032011). We propose a similar mechanism in our system in the inhomogeneous presence of an additional weak enzyme (CytC). The random rotation of the nanomotor chassis presumably diminishes due to the CytC-induced active rotations towards the pyrogallol gradient despite the weak catalytic proficiency. This installs a weak external field which eventually restricts the random rotations of the urease+CytC-powered self-diffusiophoretic motors biasing the direction of motion towards the higher concentration region of pyrogallol (*Biochemistry* **2018**, *57*, 6256–6263). In other words, this additional interaction (CytC-pyrogallol) disrupts the overall randomness in directionality in the strong persistent motion powered by urease and subsequently, the motors now make a gradual choice towards the external bias of the gradient of the CytC-substrate (*PLoS Biol.*, **2016**, *14*, e1002463, *EPL* **2013**, *103*, 60009). However, after rotation, there would still be an equal possibility of the motors to go against or towards the gradient. We argue that the choice towards the pyrogallol gradient could partially be attributed to the weak yet finite diffusivity of the CytC (CytC-loaded nanotubes in the absence of urease indeed showed enhanced diffusivity (at 100 mM pyrogallol), although the diffusivity was significantly lesser compared to urease-CytC loaded nanotubes, $D_{urease/cytc} = 6.99 \pm 0.05 \mu\text{m}^2/\text{s}$, $D_{cytc} = 0.42 \pm 0.02 \mu\text{m}^2/\text{s}$ in buffer). However, understanding the interplay between the orthogonal enzymes, that breaks the symmetry of equal possibility of moving against or towards the pyrogallol gradient, requires further dedicated investigations and our current efforts are towards this front.

Minor issues: - The phrase “nanotube-bound urease” is misleading, and should be changed to “urease -bound nanotube”.

>> We have now rephrased it in the revised SI.

I don't understand what this result means: "Notably, rapid generation of purpurogallin was observed in the gel buffer interface with a 7-fold higher rate than the controls without urea (Supplementary Figs. 20, 23)"

>> We have now elaborated the discussion (page 10, yellow highlight, Revised MS). Briefly, in this experiment, the generation of the purpurogallin that was released in the gel-buffer interphase was monitored with time. We observed that the oxidation of pyrogallol was 7-fold higher when urea was present in the medium in the presence of the urease-CytC nanomotors. This result suggests that the chemotaxis towards higher pyrogallol concentrations at the interphase led to the higher catalytic activity of CytC (as substrate concentration was more). While in the absence of urea, the nanotubes showed a random distribution of motion and did not lead to specific localization to higher pyrogallol concentration (gel-buffer interface) leading to lower catalytic rates of CytC (as pyrogallol concentration was low).

The description of the following experiment is misleading “Different concentrations of pyrogallol (0 mM, 50 mM, 100 mM) loaded glass capillaries...”. The main text only mentions “enzyme-loaded nanomotors and H₂O₂”, which to me suggested that the motor was functionalized solely with ctyc and no urease or urea was present in this experiment. This would be an interesting experiment to show that ctyc alone could cause chemotaxis. However, I later found in Supporting Information that “Dual enzyme-loaded nanomotors along with urea were added to the buffer system”. This is misleading to say the least. With this new knowledge, I can't help but wonder what is the point of performing this set of experiment, because it is basically a repetition of the results in Fig. 4

>> This was an oversight; we thank the reviewer for pointing this out. We have now corrected the statement “enzyme-loaded nanomotors and H₂O₂” to “urease-CytC loaded nanomotors and H₂O₂.” (page 11, yellow highlight, Revised MS).

As suggested by the Reviewer, the chemotactic ability of only CytC-loaded nanotubes was investigated using a similar setup. We monitored the population at the opening of the capillary. A modest increase was observed after a prolonged time of ca. 1.5h in the overall population of the nanotubes (Supplementary Fig. 27b, Revised SI). It would be important to mention here that in presence of urea, the urease-CytC nanotubes showed localization in ca. 15-20 min. In combination, these results suggest that motors in absence of urea do have a subdued yet finite chemotactic propensity towards a gradient of pyrogallol and it takes longer time to achieve this. As noted earlier in the response, CytC-loaded nanotubes in the absence of urease indeed showed enhanced diffusivity (at 100 mM pyrogallol), although the diffusivity was significantly lesser compared to urease-loaded nanotubes ($D_{urease/ctyc} = 6.99 \pm 0.051 \mu\text{m}^2/\text{s}$, $D_{ctyc} = 0.42 \pm 0.017 \mu\text{m}^2/\text{s}$). However, when urea was included in the medium (for the capillary-experiment mentioned above), the localization of the nanotubes increased significantly after 15-20 min (Supplementary Fig. 28, Revised SI).

Reviewers' Comments:

Reviewer #4:

Remarks to the Author:

I'm reviewing this article again upon the invitation of the Editor, who has asked me to evaluate the authors' response to Reviewer #1.

My evaluation is that the authors have done their best to address the reviewer's comments and suggestions. These two parties differ on what is required to publish this work. The reviewer has insisted that a reasonable mechanism to be provided for the observed chemotaxis, while the authors believe they have done enough. The reviewer has also seemingly neglected a few things the authors have provided in their revision.

Although it would be very important to unravel the cause of chemotaxis, in practice this is quite challenging, and few (if any) of the reported colloidal chemotaxis in synthetic systems have been 100% clarified. Taking a step back, I would recommend the publication of an experimental study if it can provide strong evidence that chemotaxis does occur, and if it proposes a plausible mechanism. My opinion is that after several revisions this manuscript has fulfilled this requirement. A detailed description of how it has done so is given below in my original review.

I therefore recommend the publication of the revised manuscript, provided the minor issues listed below are addressed properly.

I applaud the authors' extensive efforts in addressing all of my comments. In particular, they have provided a complete description of all the control experiments, which very nicely resolved all my doubts. They have also performed an additional experiment where active and passive rods were mixed, as I suggested. The result was in nice agreement with expectations. Moreover, they have made an effort in understanding the mechanism of chemotaxis. Although this mechanism is still far from being perfect, I do believe that significantly more effort is needed to clarify the mechanism and it is beyond the scope of this work.

Overall, I have no objection for the publication of this work, if the following minor issues can be addressed in a revision (I do not need to review it again).

1. Please provide a supporting video of the self-propulsion of nanotubes that are not functionalized with urease but in the presence of urea. This video will be useful in proving that the motion seen in Video S1-4 is not due to convection.
2. I do not understand the phrase "orthogonal substrate scopes" in the abstract.
3. Page 188 on page 10, I do not understand "CytC bound on nanotubes". Does it refer to the nanotubes bound with cytc?
4. In fig. 1b, I suggest making a connection between the pink circle with the big arrow below it, so that a reader knows the pink arrows in the circle is made of the molecule below.

Reviewer #4 (evaluation of response to Reviewer #1's comment):

I'm reviewing this article again upon the invitation of the Editor, who has asked me to evaluate the authors' response to Reviewer #1.

My evaluation is that the authors have done their best to address the reviewer's comments and suggestions. These two parties differ on what is required to publish this work. The reviewer has insisted that a reasonable mechanism to be provided for the observed chemotaxis, while the authors believe they have done enough. The reviewer has also seemingly neglected a few things the authors have provided in their revision.

Although it would be very important to unravel the cause of chemotaxis, in practice this is quite challenging, and few (if any) of the reported colloidal chemotaxis in synthetic systems have been 100% clarified. Taking a step back, I would recommend the publication of an experimental study if it can provide strong evidence that chemotaxis does occur, and if it proposes a plausible mechanism. My opinion is that after several revisions this manuscript has fulfilled this requirement. A detailed description of how it has done so is given below in my original review.

I therefore recommend the publication of the revised manuscript, provided the minor issues listed below are addressed properly.

>> We thank the Reviewer for considering the work suitable for publication. We appreciate that the Reviewer is satisfied with our response and the analysis.

Reviewer #4

I applaud the authors' extensive efforts in addressing all of my comments. In particular, they have provided a complete description of all the control experiments, which very nicely resolved all my doubts. They have also performed an additional experiment where active and passive rods were mixed, as I suggested. The result was in nice agreement with expectations. Moreover, they have made an effort in understanding the mechanism of chemotaxis. Although this mechanism is still far from being perfect, I do believe that significantly more effort is needed to clarify the mechanism and it is beyond the scope of this work.

Overall, I have no objection for the publication of this work, if the following minor issues can be addressed in a revision (I do not need to review it again).

>> We appreciate the enthusiasm of the Reviewer for the publication of the work.

1. Please provide a supporting video of the self-propulsion of nanotubes that are not functionalized with urease but in the presence of urea. This video will be useful in proving that the motion seen in Video S1-4 is not due to convection.

>> The video of the nanotubes that are not functionalized with urease in the presence of urea is included as (Supplementary Video 5). Expectedly, the nanotubes did not show any enhanced motility in the absence of the powering source urease.

2. I do not understand the phrase "orthogonal substrate scopes" in the abstract.

>> By the phrase 'orthogonal substrate scopes', we meant that the two enzymes work on completely different substrates. We have simplified the sentence "...to catalyze orthogonal substrates for motility and ..."

3. Page 188 on page 10, I do not understand "CytC bound on nanotubes". Does it refer to the nanotubes bound with cytc?

>> The phrase "CytC bound on nanotubes" refers to the nanotubes loaded with CytC.

4. In fig. 1b, I suggest making a connection between the pink circle with the big arrow below it, so that a reader knows the pink arrows in the circle is made of the molecule below.
>> As suggested by the Reviewer, we have now revised Figure 1b (revised MS).